# Deep-profiling of phospholipidome via rapid orthogonal separations and isomer-resolved mass spectrometry

Tian Xia[1], Feng Zhou[2], Donghui Zhang[3], Xue Jin [4], Hengxue Shi[1], Hang Yin [4,5,6], Yanqing Gong[7] & Yu Xia [1] ✉

A lipidome comprises thousands of lipid species, many of which are isomers and isobars. Liquid chromatography-tandem mass spectrometry (LC-MS/MS), although widely used for lipidomic profiling, faces challenges in differentiating lipid isomers. Herein, we address this issue by leveraging the orthogonal separation capabilities of hydrophilic interaction liquid chromatography (HILIC) and trapped ion mobility spectrometry (TIMS). We further integrate isomer-resolved MS/MS methods onto HILIC-TIMS, which enable pinpointing double bond locations in phospholipids and sn-positions in phosphatidylcholine. This system profiles phospholipids at multiple structural levels with short analysis time (<10 min per LC run), high sensitivity (nM detection limit), and wide coverage, while data analysis is streamlined using a home-developed software, LipidNovelist. Notably, compared to our previous report, the system doubles the coverage of phospholipids in bovine liver and reveals uncanonical desaturation pathways in RAW 264.7 macrophages. Relative quantitation of the double bond location isomers of phospholipids and the sn-position isomers of phosphatidylcholine enables the phenotyping of human bladder cancer tissue relative to normal control, which would be otherwise indistinguishable by traditional profiling methods. Our research offers a comprehensive solution for lipidomic profiling and highlights the critical role of isomer analysis in studying lipid metabolism in both healthy and diseased states.

A lipidome typically contains thousands of lipid molecular species that includes isomers and isobars, with concentrations spanning six to eight orders of magnitude[1,2]. In the mammalian lipidome, lipids exhibit high structural diversity to support multifaceted functions involving cell structuring[2], signaling[3], and energy storage[4]. Lipid profiling via mass spectrometry (MS) is routinely employed in lipidomic studies as it provides a direct readout of lipid metabolism of a given biological state. Typically, lipid profiling is achieved via either direct infusion electrospray ionization (ESI), termed as shotgun lipid analysis[5], or via coupling liquid chromatography (LC) with ESI-MS[6–8]. The use of LC effectively mitigates ion suppression and isobaric interferences encountered in direct infusion methods. Additionally, dedicated LC methods have been developed to separate lipid isomers with variations in carbon−carbon double bond (C = C) position and geometry[9],

[1]MOE Key Laboratory of Bioorganic Phosphorus Chemistry & Chemical Biology, Department of Chemistry, Tsinghua University, 100084 Beijing, China. [2]Bytedance Technology Co., 201103 Shanghai, China. [3]State Key Laboratory of Precision Measurement Technology and Instruments, Tsinghua University, Department of Precision Instrument, 100084 Beijing, China. [4]School of Pharmaceutical Sciences, Tsinghua University, 100084 Beijing, China. [5]Tsinghua-Peking Center for Life Sciences, Tsinghua University, 100084 Beijing, China. [6]Beijing Frontier Research Center for Biological Structure, Tsinghua University, 100084 Beijing, China. [7]Department of Urology, Peking University First Hospital, 100034 Beijing, China. ✉e-mail: xiayu@mail.tsinghua.edu.cn

as well as lipid phosphate regio-isomers[10,11] and enantiomers[12], albeit requiring moderate to long separation time (20 min to several hours). Alternatively, ion mobility spectrometry (IMS) enables fast gas-phase separation of molecules according to their shape, size, and charge on millisecond (ms) timescales. High-resolution IMS techniques when coupled with LC-MS thus bring distinct advantages in lipid profiling from complex mixtures[13–16]. Blaženović et al. achieved improved predictions for unknown lipid features by including collision cross section (CCS) values in their compound identification model[17]. Lerner et al. demonstrated high-throughput profiling of lipids on an MS platform consisting of microflow reversed-phase liquid chromatography (RPLC), trapped ion mobility spectrometry (TIMS), and parallel accumulation-serial fragmentation[18]. The utilization of four-dimensional data, including $m/z$, tandem mass spectra, LC retention time, and CCS values, enables the application of stringent criteria for lipid annotation, consequently reducing the extent of misidentifications, a crucial aspect that has gained considerable attention within the lipidomics community[19]. However, these cutting-edge MS platforms can only profile lipids at the sum composition level or fatty acyl chain level; detailed levels of structural information are missing, including C = C location and the relative position of the fatty acyl/ether chain on a glycerol backbone (*sn*-position).

Lipid profiling at detailed structural levels, or so called deep-profiling, requires the use of isomer-resolved tandem mass spectrometry (MS/MS) techniques[20]. New MS/MS methods have been developed for locating C = C in different classes of lipids. These include ozone-induced dissociation (OzID)[21], ultraviolet photodissociation (UVPD)[22], electron impact excitation of ions from organics (EIEIO)[23], ion/ion reactions[24], and oxygen attachment dissociation[25]. Alternatively, C = C derivatization methods have shown success when coupled with MS/MS, such as the Paternò-Büchi (PB) reaction[26], epoxidation[27–30], singlet oxygen derivatization[31], and aziridination[32,33]. Our group and others have shown that the PB derivatization approach is MS instrument independent and it can be incorporated into different lipidomic settings, including shotgun[34,35], LC-MS[36,37], direct analysis[38,39], and MS imaging[40,41]. Overall, the new capability of lipidome-wide profiling of C=C location isomers has enabled the discovery of altered lipid metabolism in diseases which would be otherwise overlooked by just employing traditional lipid profiling workflows[16,34,37,42–45].

An ideal lipid profiling tool should not only provide accurate identification of lipid molecules at detailed structural levels, but also be feasible for large-scale quantitative analysis. In an earlier study, we developed a hydrophilic interaction liquid chromatography (HILIC)-PB-MS/MS workflow in which acetone PB reaction was installed inline after HILIC separation and right before ESI-MS[36]. This workflow made it possible to conduct lipidome-wide comparisons of compositional changes of C=C location isomers for disease phenotyping. As powerful as it was, we also observed several limitations. Firstly, the same class of phospholipids which typically encompasses more than 100 detectable $m/z$ values, coelute within 1–2 min on HILIC, leading to notable ion suppression for lower abundance lipids of the same class. Secondly, the +2 Da isotope interference caused by the lipid analog of one more degree of unsaturation is unavoidable in HILIC-MS, which brings uncertainties in C=C identification as well as isomer quantitation. Thirdly, the mass envelope of acetone PB reaction products typically overlaps with that of the unreacted lipids of the same class due to the small mass increase (58 Da). This effect also has an unfavorable effect on collecting high-quality PB-MS/MS data. Alternatively, the newly developed PB reagents, such as 2′,4′,6′-tri-fluoroacetophenone (triFAP, 174 Da) can effectively separate the two populations of ions to enhance analysis of low abundance lipids[46]. To address the above issues and keep the advantages of HILIC, we believe the best solution is to couple HILIC separations with high-resolution IMS, which offers orthogonal separation capabilities in a millisecond timescale.

Herein, we have integrated HILIC, IMS, and isomer-resolved MS/MS methods into a single workflow, aiming to achieve large-scale, fast, and sensitive analysis of lipids at detailed structural levels. The workflow features a relatively short HILIC separation (<10 min per run) for eight major classes of glycerophospholipids (GPLs) and sphingomyelins (SMs), with significantly improved capability for reducing isobaric interference via coupling with TIMS separations. The employment of offline triFAP PB reaction and MS² CID of the bicarbonate anion adduct of phosphatidylcholine (PC) allows profiling phospholipids at the C = C location level as well as mapping the distribution of *sn*-position isomers of phosphatidylcholine (PC), respectively. The data are analyzed by a home-built tool, LipidNovelist, which supports automatic lipid structural annotation. The analytical capability of this work is manifested by deep profiling of phospholipids from the polar extract of bovine liver, RAW 264.7 macrophages, and human bladder cancer tissue samples.

## Results

### HILIC-TIMS-MS/MS for detailed structural analysis of GPLs

Different classes of GPLs all share a phosphodiester functional group while differing in the identity of the headgroup. This makes GPLs ionize reasonably well in negative ion mode via ESI except for PCs which contain a fixed positive charge in the choline moiety[47]. Our earlier work has shown that the addition of $NH_4HCO_3$ to the LC mobile phase led to at least 10 times enhanced detection of PC and SM via forming a relatively strong bicarbonate anion adduct ($[M + HCO_3]^-$) than the commonly used acetate or formate anion adduct[43,48]. MS² CID of $[PC + HCO_3]^-$ forms *sn*-1 specific fragment ions which provides identification and quantitation of *sn*-position isomers[43]. Therefore, performing ionization and MS² CID in negative ion mode is advantageous for the analysis of GPLs at the fatty acyl composition level. We then evaluated the impact of charge polarity on separations via TIMS using a mixture of GPL standards. Under the same scan rate (1 V/ms) of TIMS, better separations were achieved for the anionic ions of different GPL standards, with an average resolving power of 86 as compared to 52 for the positively charged ions, likely due to forming more compact gas-phase conformation than corresponding cationic forms (Fig. 1a)[14].

HILIC separates GPLs by headgroups; however, it cannot separate the same class of lipids differing in one degree of unsaturation, which causes type II chemical interferences for identification and quantitation[49]. For instance, PC 34:1 is more abundant than PC 34:0 in mammalian lipidome and they coelute on HILIC (Supplementary Fig. 1). MS¹ of a standard mixture of PC 16:0/18:1 (1 μM) and PC 18:0/16:0 (0.1 μM) thus shows a significant overlap of the monoisotope of PC 18:0/16:0 and the +2 Da isotope of PC 16:0/18:1 at $m/z$ 822.585 ($[M + HCO_3]^-$, Fig. 1b). Previously, we have established a quantitative analysis method for the *sn*-isomers of PC by detecting the *sn*-1 position specific ions, termed as "*sn*-1 fragment", formed in MS² CID of $[M + HCO_3]^{-43}$. For instance, MS² CID of pure PC 18:0/16:0 only produces *sn*-1 18:0 ion ($m/z$ 447.291, structure shown in the inset of Fig. 1), while that related to *sn*-2 16:0 should not be formed. Because the commercial standard of PC 18:0/16:0 consists of its *sn*-isomer, PC 16:0/18:0, as a minor impurity, a small peak corresponding to the *sn*-1 16:0 fragment ($m/z$ 419) is also observed (Supplementary Fig. 2). The purity of PC 18:0/16:0 is calculated to be 82.0 ± 0.7% by normalizing the ion abundance of *sn*-1 18:0 to the sum of *sn*-1 18:0 and *sn*-1 16:0 fragments in the same spectrum. The accuracy of quantitation using the *sn*-1 fragment has been verified by the phospholipase A2 assay previously[43]. For the situation in Fig. 1b, where the contribution of +2 Da isotope of PC 16:0/18:1 is significant, the *sn*-1 16:0 fragment ($m/z$ 419, Fig. 1c) is more abundant than it should be, thus leading to a lower %purity of PC 18:0/16:0 (66 ± 2% as compared to 82.0 ± 0.7%).

By using TIMS, the +2 Da isobar of PC 34:1 can be just partially separated from the monoisotopic peak of PC 34:0 using a scan rate of

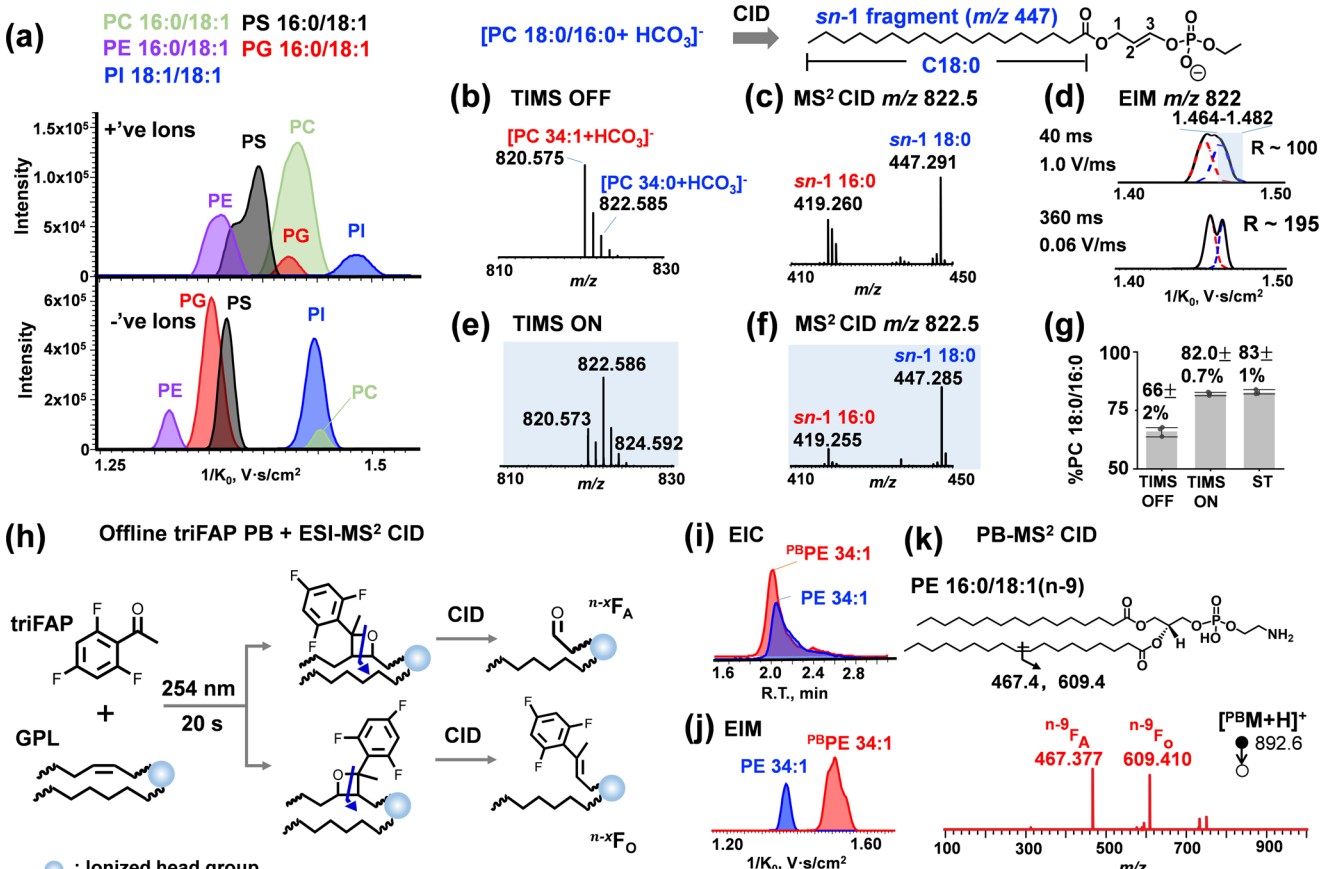

**Fig. 1 | HILIC-TIMS-MS/MS system for detailed structural analysis of GPLs.**
**a** Extracted ion mobilograms (EIMs) of five GPL standards detected in positive ion mode ([M + H]⁺ for PC, PE, and PS, [M + NH₄]⁺ for PG and PI, upper panel) and negative ion mode ([M−H]⁻ for PG, PI, PS, and PE, [M + HCO₃]⁻ for PC, lower panel). **b** Negative ion mode MS¹ spectrum of a 10:1 mixture of PC 16:0/18:1 and PC 18:0/16:0, detected as [M + HCO₃]⁻. **c** Zoomed-in MS² CID spectrum of *m/z* 822.59 formed in panel **b** without TIMS separation. Only the *m/z* region of *sn*-1 fragment is shown. Inset shows the structure of the *sn*-1 18:0 fragment (*m/z* 447) derived from MS² CID of PC 18:0/16:0 ([M + HCO₃]⁻). **d** EIMs of *m/z* 822.585 (black trace) formed in panel **b** with a scan rate of 1 V/ms and 0.06 V/ms during TIMS separation. The +2 Da isotope of PC 16:0/18:1 and the mono-isotope of PC 18:0/16:0 are gaussian

deconvoluted and shaded in red and blue, respectively. Resolving power (R) is listed. Mobility-resolved (**e**) MS¹ spectrum and (**f**) MS/MS spectrum of *m/z* 822.59 isolated from 1/K₀ of 1.464–1.482 V·s/cm² in panel **d**. **g** Comparison of the purity of PC 18:0/16:0 without and with TIMS separation (N = 3 independent experiments). Bars represent mean values ± standard deviation (SD). (**h**) Schematic presentation of triFAP PB-MS² CID for generating C = C diagnostic ions. (**i**) Extracted ion chromatograms (EICs) of unreacted PE 16:0/18:1(n-9) (blue trace) and triFAP-modified PE 16:0/18:1(n-9) (red trace). **j** EIMs of unreacted PE 16:0/18:1(n-9) (blue trace) and triFAP-modified PE 16:0/18:1 (n-9) (red trace). **k** MS² CID spectrum of triFAP-derivatized PE 16:0/18:1(n-9) ([ᴾᴮM + H] ⁺, *m/z* 892.6). Source data are provided as a Source Data file.

1 V/ms (R ~ 100, Fig. 1d upper panel), in which the mobility range of 1.464–1.482 (1/K₀, V·s/cm²) (blue rectangle) is contributed by PC 34:0 as a major component. The corresponding mobility selected MS¹ spectrum (Fig. 1e) thus shows the correct isotopic pattern of PC 18:0/ 16:0 with only a mitigated amount of PC 34:1 alongside. MS² CID of *m/z* 822.585 ([M + HCO₃]⁻) after TIMS separation produces a much cleaner MS² spectrum as shown in Fig. 1f. The purity of PC 18:0/16:0 was calculated to be 83 ± 1%, very similar to that obtained from a separate analysis of the PC 18:0/16:0 standard (Fig. 1g). Although better separation of the isobars can be achieved by using a scan rate of 0.06 V/ms (R ~ 195, Fig. 1d lower panel), it takes much longer scan time (360 ms vs. 40 ms), thus reducing the throughput of MS/MS analysis. The above set of experiments demonstrate that TIMS complements HILIC by providing reasonable separations of co-eluting lipids and improves quantitation at the isomer level. It should be noted that the CCS values of the *sn*-isomers[50] of most phospholipids are very close to each other and it requires a resolving power (>500)[51] much higher than that can be achieved on the current TIMS module. This again highlights the importance of using isomer-resolved MS/MS to achieve quantitation especially when the isomers cannot be separated.

The analysis of GPLs at the chain composition level was conducted via MS² CID in negative ion mode. The deprotonated ions ([M−H]⁻) were the predominant ionic form for all classes of phospholipids except for lysophosphatidylcholine (LPC), PC, and SM, for which the bicarbonate anion adducts ([M + HCO₃]⁻) were formed. Due to the peak concentrating effect in TIMS, at least 20-fold increases in signal-to-noise ratio (S/N) were achieved relative to continuous acquisition mode (TIMS off). With TIMS on, the limit of detection (LOD) at the chain composition level for PI 18:1/18:1, PG 16:0/18:1, and PE 16:0/18:1 was around 0.1 nM, while 1 nM LOD was achieved for PC 16:0/18:1, and PS 16:0/18:1. Overall, combing HILIC with TIMS-MS/MS produces high-quality MS/MS spectra for confident structural identification and quantitation at the chain composition level.

As an efficient PB reagent, triFAP offers sensitive identification as low as nM for GPLs on the C = C location level when coupled with MS/MS in positive ion mode. Note that PB-MS² CID of phospholipids in negative ion mode is much less sensitive in providing detailed structural information. A generic PB reaction scheme, as well as the structures of C = C diagnostic ions (ⁿ⁻ˣFₐ and ⁿ⁻ˣF_O) formed from MS² CID, is shown in Fig. 1h. The subscript "A" or "O" indicates an aldehyde or an olefin moiety at the cleavage site in the fragment, while the

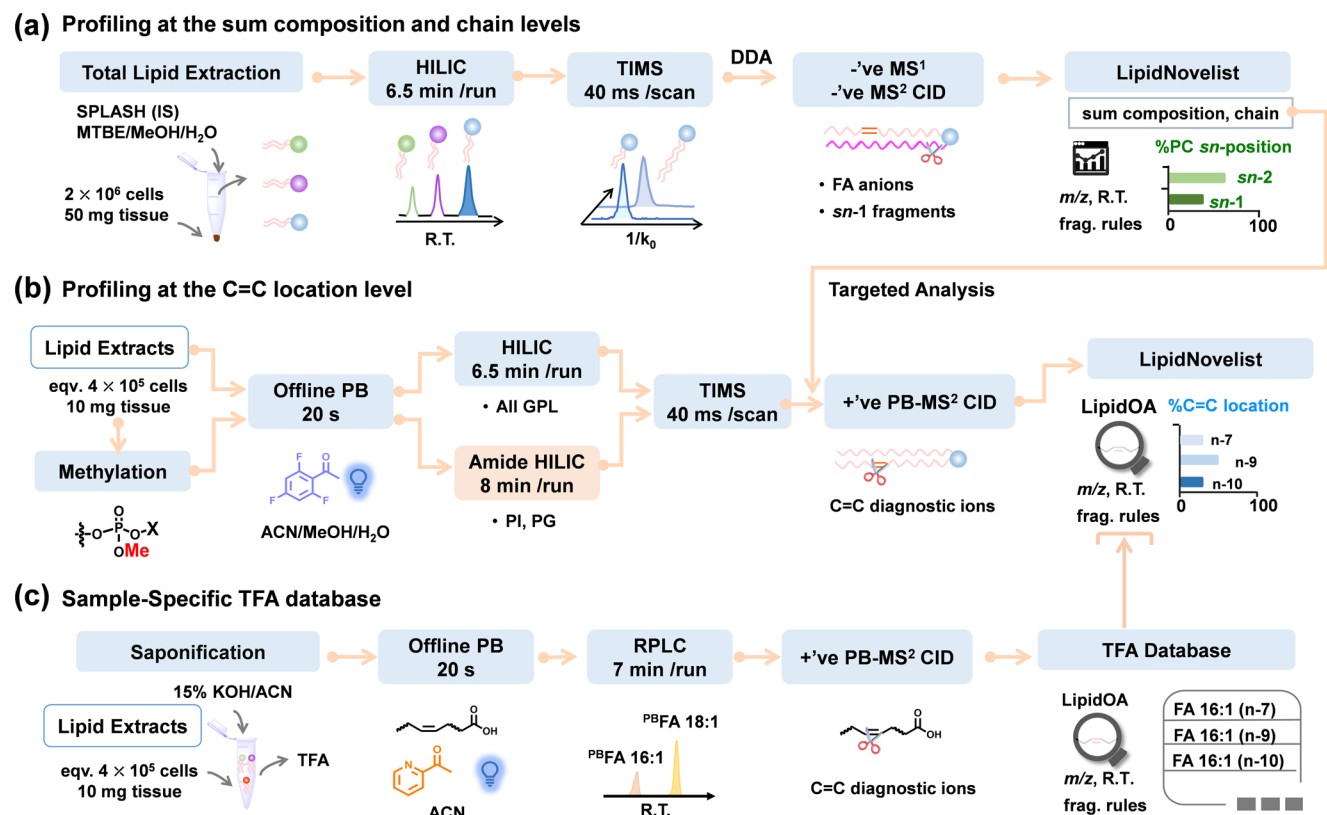

**Fig. 2 | The workflow for deep profiling of phospholipids. a** HILIC-TIMS-MS/MS in negative ion mode via DDA to profile lipids at the sum composition, chain composition, and *sn*-position (for PC only) levels. **b** Offline PB reaction coupled with HILIC-TIMS-MS/MS in positive ion mode via targeted analysis to profile lipids at the level of C = C location. To enhance detection sensitivity for PI and PG, the lipid extracts are subjected to phosphate methylation before subsequent amide HILIC-TIMS-MS/MS. **c** Sample-specific TFA database is established via coupling an offline PB reaction with RPLC-MS/MS. A home-built data analysis package, LipidNovelist, is used to guide data collection and perform structural annotation and relative quantitation at different structural levels.

superscript, n-x, denotes the location of the C = C following the n-nomenclature. PG and PI were subjected to phosphate methylation before triFAP derivatization to improve the ionization in positive ion mode (Supplementary Fig. 3)[37]. We thus paired 20 s offline triFAP PB reactions with HILIC-TIMS-MS/MS (kinetic studies of the PB reactions provided in Supplementary Fig. 4a). Using PE 16:0/18:1(n-9) as an example, the triFAP-derivatized lipid coelutes with the unreacted lipid on a HILIC column (Fig. 1i); these two species, however, can be well separated by TIMS (Fig. 1j). Thus, the addition of TIMS for the PB products is expected to reduce chemical interference and improve detection limits for low abundance lipids. It should also be noted that the peak width of triFAP-derivatized PE is widened by about 1.6-folds as compared to the remaining PE in the mobilogram. This phenomenon is consistently observed for polyunsaturated lipids such as PE 38:3 and PE 38:4 (Supplementary Table 1). Consequently, in this study TIMS could not separate the type II interference for triFAP-modified GPLs (Supplementary Fig. 4b). Figure 1k shows the MS[2] CID spectrum of [$^{PB}$PE 16:0/18:1(n-9) + H]$^+$ (*m/z* 892.57). Abundant C = C diagnostic ions are detected at *m/z* 467.38 ($^{n-9}F_A$) and *m/z* 609.41 ($^{n-9}F_O$). The mass difference between the pair of diagnostic ions from the same C = C location is 142 Da, characteristic of triFAP PB-MS/MS. The sensitivity of PB-MS/MS was evaluated by serial dilution with on-sample LODs achieved at 10 nM for PC 16:0/18:1(n-9) and PS 16:0/18:1(n-9), 2.5 nM for PE 16:0/18:1(n-9), PG$^{Me}$ 16:0/18:1(n-9), and PI$^{Me}$ 18:1/18:1(n-9), comparable to the previously reported LC-PB-MS/MS workflow in which the acetone PB reaction was implemented online[36,37].

## HILIC-TIMS-MS/MS system for deep-profiling of phospholipids

The typical chromatographic peak widths for individual phospholipids on HILIC are in the range of 2–4 s, while the TIMS analysis is 40 ms at

1 V/ms scan rate per precursor ion mass, and the acquisition speed of TOF for MS[1] and MS/MS is about 100 μs per spectrum. Therefore, an integration of HILIC, TIMS, and isomer-resolved MS/MS into a single analysis run has distinct analytical advantages regarding the speed, sensitivity, and confidence in lipid identification. Figure 2 summarizes such a workflow for deep profiling of phospholipids. It comprises three key steps for data acquisition and a home-built data analysis package, LipidNovelist, to guide data collection and perform structural annotation and relative quantitation at different structural levels (Supplementary Note 1). As shown in Fig. 2a, the first step involves data-dependent analysis (DDA) in negative ion mode via a 6.5 min HILIC-TIMS-MS run (5 min separation and 1.5 min re-equilibration). The goal is to achieve relative quantitation at the sum composition level, obtain chain information of all detected phospholipids, and analyze the *sn*-position of PC. Under the LC-MS condition used, deprotonated ions ([M−H]$^-$) are the predominant ionic form for all classes of phospholipids except for LPC, PC, and SM. For the latter ones, the bicarbonate anion adducts [M + HCO$_3$]$^-$ are formed abundantly. Each DDA cycle is 0.2 s, comprising the acquisition of one TIMS-MS[1] survey spectrum and four TIMS-MS[2] CID spectra by fragmenting the N (N = 4) most abundant precursor ions. On average, precursor ions are fragmented three times and dynamically excluded for 0.1 min. Lower abundance precursor ions are repeatedly fragmented to increase their signal-to-noise ratios in a summed spectrum. This non-targeted acquisition provides over 5000 MS[2] CID in each LC run covering eight major classes of phospholipids. For the acquired MS[1] data, LipidNovelist Extension performs relative quantification (I/I$_{IS}$) at the sum composition level after Type-I isotope correction[52]. For the MS[2] CID data, LipidNovelist provides annotations of phospholipids with specific chain information according to class-specific retention time (R.T.), accurate *m/z*, and

fragmentation rules. Additionally, when analyzing the MS$^2$ CID spectra of [PC + HCO$_3$]$^-$, LipidNovelist identifies the *sn*-information of the PC species and calculates the %composition of the *sn*-isomers using the ion abundance of *sn*-1 fragment.

To profile phospholipids at the C = C location level, the lipid extract is first derivatized by triFAP via offline PB reaction. LipidNovelist generates a list of precursor ions from the first run which corresponded to the triFAP-derivatized unsaturated lipids ([M + H]$^+$ for LPC, LPE, PC, PE, and PS, and [M + NH$_4$]$^+$ for methylated PG and PI). The targeted analysis is applied to the PB derivatized lipids via a 6.5 min HILIC-TIMS-MS/MS run (Fig. 2b). To enhance sensitivity for anionic lipids such as PG and PI, the lipid extract is subjected to a separate phosphate methylation procedure[37] before triFAP derivatization and the amide HILIC-TIMS-MS/MS analysis. The PB-MS/MS data are analyzed by the LipidOA module[53] embedded in LipidNovelist to annotate phospholipids at the C = C location level. Although the C = C location can be determined in phospholipids with *sn*-specific information[21,22,34], it requires multiple-stage MS$^n$ (n ≥ 3) which results in lower sensitivity and throughput. Therefore, we did not pursue merging the assignment of *sn*- and C = C location for each identified phospholipid in this workflow.

A sample-specific total fatty acid (TFA) database with known C = C location information is created via a procedure shown in Fig. 2c[44]. The TFA of a specific lipid extract is obtained via saponification and subsequently derivatized by charge-tagging PB reaction and analyzed by an RPLC-MS/MS run in positive ion mode. LipidOA analyzes the PB-MS/MS data and generates the TFA database with C = C location information. The database is further used by LipidOA as prior-known knowledge to improve accuracy during de novo annotation of phospholipids in Fig. 2b. It should be noted that if a TFA database has been established before, this step can be omitted.

For benchmark purpose, the above system was applied to analyze the polar lipid extract of bovine liver (1 µg lipid extracts per injection), a commercially available lipid mixture with phospholipids being extensively analyzed at the C = C location level using different PB-MS/MS methods[36,46]. The performance of the system was further evaluated for profiling unknown phospholipidomes using RAW 264.7 cells (~20,000 cells per injection). A significant aspect of this study was the development of a sample-specific TFA database by correlating desaturase inhibition with the compositional changes of C = C isomers in TFAs. This approach overcame the practical limitation of obtaining lipid C = C isomer standards and significantly improved the confidence in phospholipid annotation. Finally, to demonstrate the unique capability of deep profiling in clinical applications, the human bladder tissue samples were analyzed (100 µg tissue per injection).

### Analysis of phospholipids in bovine liver polar lipid extract

The performance of the HILIC-TIMS-MS/MS system was evaluated using the polar lipid extract of bovine liver (1 µg lipid per injection). As shown in Fig. 3a, most classes of GPLs are well separated by HILIC, except for PI and PG. This issue is resolved by TIMS, in which PI and PG are readily separated due to large structural differences between the two types of headgroups (Supplementary Fig. 5). Lipids of the same class but differing in acyl chain lengths or degrees of unsaturation can be separated by TIMS (Supplementary Figs. 6 and 7). TIMS also provides a certain degree of separation of isobaric lipids, such as diacyl-PE and ether-PE (PE-O), leading to more confident identification. As an example, PE 34:3 (*m/z* 712.4923) and PE O-35:3 (*m/z* 712.5287) can be partially resolved by the TOF mass analyzer which possesses a resolving power of around 30,000 (Fig. 3b). Nevertheless, limitations associated with the isolation window of a quadrupole mass filter result in the co-isolation of lipid isobars during MS/MS experiment, producing a mixed MS/MS spectrum. PE(O-35:3) thus could be falsely identified as a combination of PE O-17:0/18:3, PE O-17:1/18:2, and PE O-19:3/16:0 due to the detection of fatty acyl anions C16:1

(*m/z* 253.25), C16:0 (*m/z* 255.23), C18:3 (*m/z* 277.21), and C18:2 (*m/z* 279.23) (Supplementary Fig. 8). On the other hand, TIMS provides a partial separation of these two species (Fig. 3c). Subsequent mobility-resolved MS$^2$ CID produces clean data for PE 34:3 (Fig. 3d) and PE O-35:3 (Fig. 3e), respectively. The data reveal that PE 34:3 is a mixture of PE 16:0_18:3 (major) and PE 16:1_18:2 (minor), while PE O-35:3 is mainly contributed by PE O-17:1/18:2.

Owing to the untargeted nature via DDA, we discovered a rarely reported class of GPLs, glycerophosphonoethanolamine (PnE), in bovine liver polar lipid extract. PnE has a phosphite ether instead of the phosphodiester group in PE[54]. MS$^2$ CID of [PnE + H]$^+$ produces a neutral loss of 125.0254 Da (C$_2$H$_8$NO$_3$P, mass error: 9 ppm), specific to the phosphite headgroup (Fig. 3f). The fatty acyl composition for PnE was obtained from MS$^2$ CID of [PnE - H]$^-$. For instance, the peak at *m/z* 750.5 is identified as PnE 18:0_20:4, based on the detection of fatty acyl anions at *m/z* 283 (C18:0) and *m/z* 303 (C20:4) (Fig. 3g). PB-MS/MS of PnE 18:0_20:4 (*m/z* 938.8) generates four pairs of C = C diagnostic ions at *m/z* 439/581, 479/621, 519/661 and 559/701, corresponding to n-6, n-9, n-12 and n-15 C = C in the C20:4 chain, respectively (Fig. 3h). Combining the class-specific retention time, accurate *m/z*, identity of the fatty acyl chain, and C = C location information, this PnE is identified as PnE 18:0_20:4 (n-6,9,12,15) (structure shown in the inset of Fig. 3).

The use of TIMS allowed us to extract the CCS values of 228 phospholipids at the sum composition level based on the measured 1/K$_0$ values from bovine liver in negative mode (Supplementary Data 1 and Fig. 3i–k)[55]. These data are consistent with those obtained from other reports[14,18] (Supplementary Table 2). Figure 3i–k show the CCS vs. *m/z* plots of the identified lipids. Each lipid class is situated in a discrete location, reflecting a significant impact of the headgroup on ion mobility. Moreover, a longer fatty acyl chain induces a more extended structure, and an increased number of double bonds generates a more compact structure. It should be emphasized that the identified PnEs form distinct trend line in the CCS vs. *m/z* plot (Fig. 3j, k). PnEs are on average 0.5 Å$^2$ smaller than the PEs sharing the same acyl chain composition while 1 Å$^2$ larger than the corresponding isomeric PE-O (Fig. 3k). These findings further support that PnE is a distinct subclass of phospholipids and showcase that CCS values can be used as a descriptor to enhance lipid identification.

TIMS separation reduces chemical interferences, which facilitates the analysis of lower abundance GPLs and thus enlarges lipid coverage in a lipidome. For instance, a total of 421 phospholipids from eight classes of GPLs and SMs were identified for chain composition in bovine liver which is more than two times the lipids (205) identified without using TIMS (Supplementary Data 1). The number of identifications for each class of phospholipids is summarized in Fig. 3l. Clearly, a significant increase of coverage for low abundance (<1%, 90 vs. 33) and medium abundance (1–10%, 197 vs. 83) is achieved by using the TIMS function. Figure 3m shows the heatmap of identified PCs down to *sn*-positions. These include 21 pairs of *sn*-isomers, which is slightly better than the 19 pairs of PC *sn*-isomers for bovine liver polar lipid extract identified on a HILIC-MS platform before[43]. Fatty acyls at the *sn*-1 position show lower diversity than those at the *sn*-2 position, with C16:0 and C18:0 as two high-frequency groups.

To enhance chain specific C = C annotation for GPLs, the total fatty acids derived from saponification of bovine liver polar lipid extract were analyzed. The offline PB derivatization by 2-acetylpyridine (2-acpy) and RPLC-MS$^2$ CID led to the determination of unsaturated FAs ranging from C14 to C24 with one to six degrees of unsaturation and 17 groups of C = C location isomers (Supplementary Fig. 9). A full list of the identified unsaturated FAs is documented in the Supplementary Data 2. LipidNovelist compared C = C annotation in GPLs to that identified in the total fatty acid database and low probability unsaturated fatty acyls were removed. In total, we identified 424 unsaturated GPLs down to chain specific C = C locations (Fig. 3l and Supplementary Data 3), two times higher than that from the online

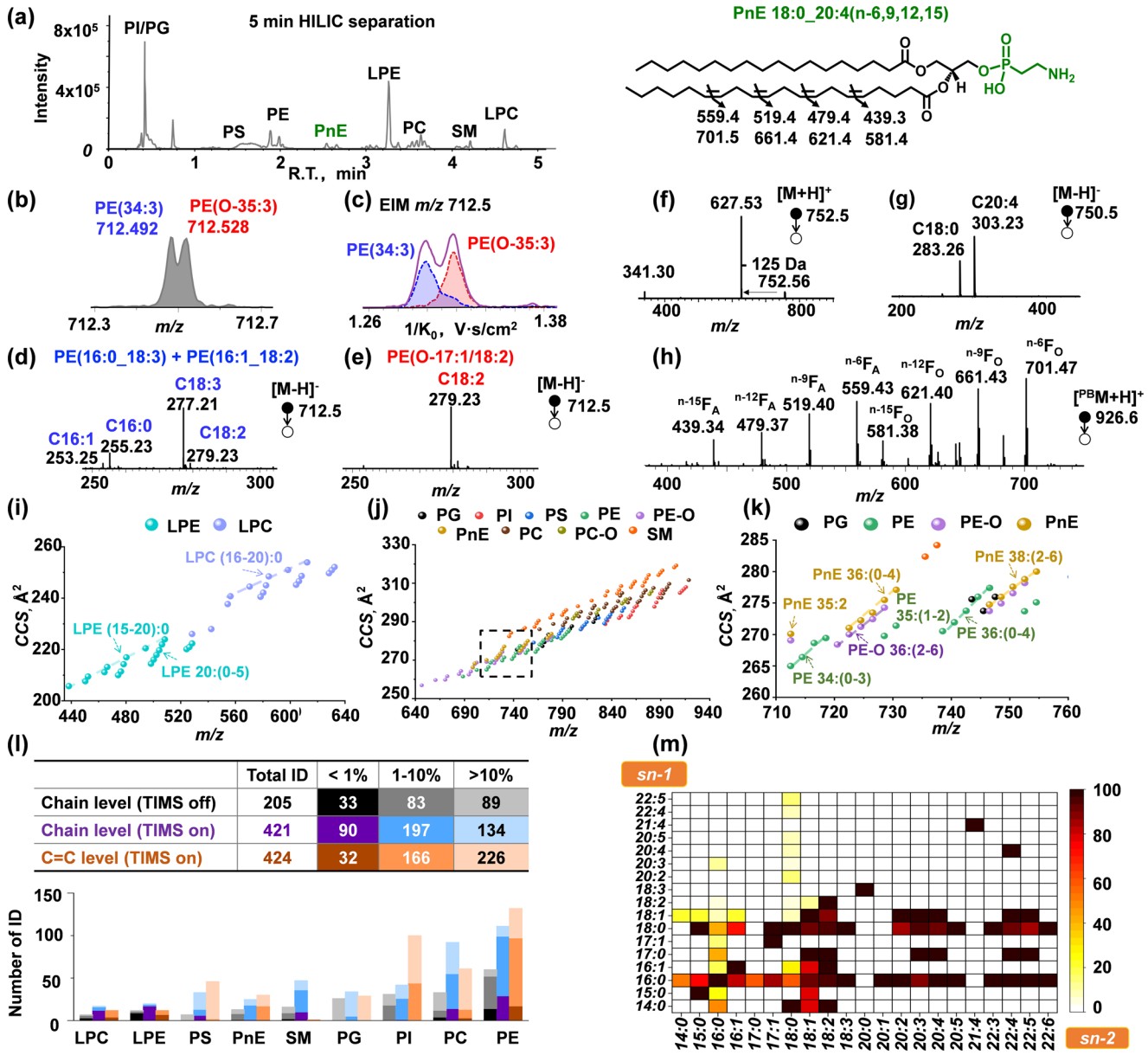

**Fig. 3 | Deep profiling of phospholipids in bovine liver. a** HILIC separation of 1 μg bovine liver polar lipid extract. **b** MS[1] spectrum of a mixture of PE 34:3 (([M -H]⁻, *m/z* 712.492) and PE O-35:3 ([M -H]⁻, *m/z* 712.528). **c** EIMs of *m/z* 712.5 in **b**. The isobaric species are gaussian deconvoluted and colored in red (PE O-35:3) and blue (PE 34:3). Mobility-resolved MS² CID spectra of *m/z* 712.5 from (**d**) 1/K₀ :1.28−1.31 and (**e**) 1/K₀ :1.31−1.34. Mass spectra of PnE 38:4 via HILIC-TIMS-MS² CID in (**f**) positive and (**g**) negative ion mode and (**h**) HILIC-TIMS-PB-MS² CID in positive ion mode. The CCS vs. *m/z* plots of 228 phospholipids identified from bovine liver in negative ion mode: (**i**) the region for LPC and LPE, (**j**) the region for GPLs and SM, and (**k**) an expanded view of the region illustrated in Fig. 3j, highlighting PG, PE, PE-O, and PnE. **l** Identification of phospholipids by HILIC-MS² CID, HILIC-TIMS-MS² CID, and HILIC-TIMS-PB-MS² CID, respectively. **m** Heatmap of fatty acyls at *sn*−1 and *sn*-2 positions of the identified PCs. Source data are provided as a Source Data file.

acetone PB-MS/MS system[36]. It is worth noting that the trans-double bond between C4 and C5 in the sphingoid base of SM has a much lower reactivity in PB reactions[48]. We didn't pursue identification of this double bond for SM when d18:1 was identified. Overall, the above set of experiments proves that triFAP PB coupled with HILIC-TIMS-MS/MS has a wide coverage and high sensitivity for profiling unsaturated GPLs in complex mixtures.

## Deep profiling of phospholipids in RAW 264.7 macrophage

RAW 264.7 cell line is a frequently used model system for in vitro studies of immune responses. Notably, quantitative lipidomic profiling conducted by the LIPID MAPS consortium has facilitated a thorough investigation of the composition, biosynthesis, and function of the major lipid classes present in RAW 264.7 cells[56,57]. Nevertheless, these

studies mainly focused on lipid identification and quantitation on the sum composition level. Herein, we aim to delineate the phospholipidome of RAW 264.7 cell line at the C = C location level, which has not been documented before and can serve as a valuable resource for studying the metabolism of unsaturated lipids. We first performed analysis on the total fatty acids to establish the TFA database. Twenty-nine groups of unsaturated FAs varying in chain lengths (C14-24) and degrees of unsaturation (1−6) were identified down to C = C location and quantified for the relative composition of isomers (Supplementary Fig. 10). %Relative composition was calculated by dividing relative ion abundance of the diagnostic ions of a certain C = C isomer by the summed ion abundance of those from all C = C isomers. A diverse distribution of C = C location isomers was detected in RAW 264.7 cells. FA 18:1 was found to consist of six C = C location isomers, contributed

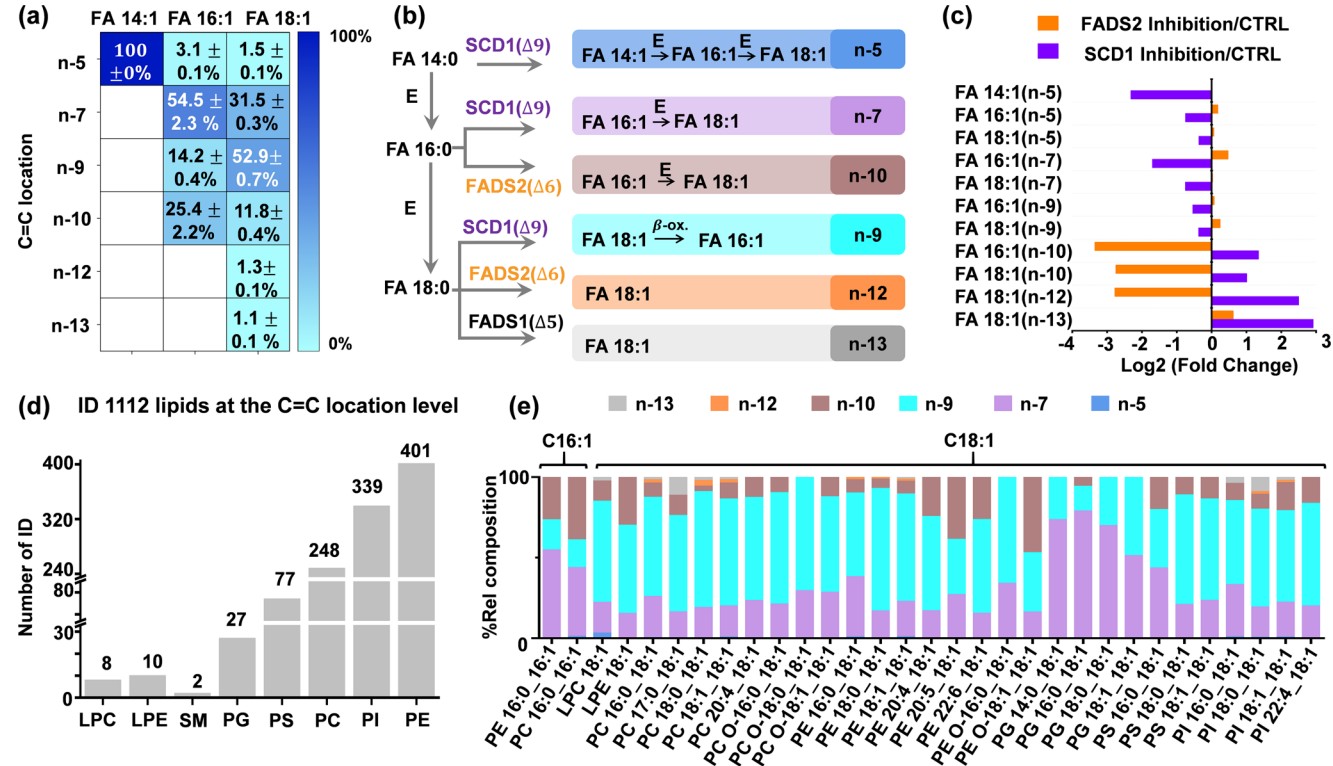

**Fig. 4 | Profiling of phospholipids at the C = C location level in RAW 264.7 macrophages. a** Heatmap of %relative compositions of C = C location isomers of FA 14:1, FA 16:1, and FA 18:1 ($N = 6$) and **b** corresponding biosynthetic pathways. E: elongation; β-ox.: β-oxidation. **c** Fold changes of C = C location isomers of 14:1, FA 16:1, FA 18:1 after inhibition of SCD-1 or FADS2 relative to the control. **d** Bar graph depicting the number of identifications for the major classes of phospholipids at the C = C level. **e** %Relative compositions of C = C location isomers for 31 groups of GPLs bearing C16:1 or C18:1 acyl chain. Source data are provided as a Source Data file.

by a predominant n-9 ($52.9 \pm 0.7\%$) and less abundant n-7 ($31.5 \pm 0.3\%$) and the n-10 isomers ($11.8 \pm 0.4\%$) (Fig. 4a). The small contribution of FA 18:1(n-5) ($1.5 \pm 0.1\%$) might result from an elongation of FA 16:1 (n-5) ($3.1 \pm 0.1\%$), which in turn is the elongation product of FA 14:1 (n-5) (100%). FA 14:1 (n-5) is likely derived from non-canonical process of SCD-1 which desaturates FA 14:0 (Fig. 4a). Young et al. recently proposed that FADS1 mediates Δ5-desaturation towards FA 18:0→FA 18:1(n-13)[42]. We also found FA 18:1 (n-13) ($1.1 \pm 0.1\%$) in RAW cell lipidome. FADS2 desaturation can yield either n-10 or n-12 FAs depending on the substrate (FA 16:0 or FA 18:0, respectively). Minor FA 18:1(n-12) ($1.3 \pm 0.1\%$) was found in RAW cells together with several other unusual FAs (Fig. 4b).

In order to have a high confidence in these unusual FA C = C location isomers especially when there are no authentic standards available, we utilized enzymatic inhibition to perturb the cells and monitored the change of C = C location isomers. The small-molecule inhibitors of FADS2 (SC26196) and SCD-1 (CAY10566) were each applied to RAW 264.7 cells. Figure 4c shows that inhibition of SCD-1 prevents the formation of n-5, n-7, and n-9 families of mono-unsaturated fatty acids by disturbing the desaturation of FA 14:0, FA 16:0, and FA 18:0 through the SCD1 pathway. This in turn consolidates desaturation activity through FADS2 and FADS1, causing increases of n-10, n-12, and n-13 isomers. On the other hand, after FADS2 inhibition relative compositions of most FADS2-related products, e.g., the n-10 and n-12 isomers, decreased significantly ($P < 0.001$, $N = 6$, Fig. 4c). FADS2 inhibition also consolidated desaturation into SCD1 pathway, rendering a prevalent increase in most n-5, n-7, and n-9 isomers (Fig. 4c). Overall, the expected compositional changes of FA C = C location isomers upon desaturase inhibition supported the identification results made by PB-MS/MS.

Due to the unexpected diversity in C=C locations discovered from the total fatty acid pool, a total of 1112 distinct structures of phospholipids were identified from RAW 264.7 cells (Fig. 4d and Supplementary Data 4), 2–3 times higher than those typically identified in mammalian tissues, such as bovine liver. We further mapped the relative composition of C = C location isomers in the C16:1–bearing and C18:1–bearing GPLs (Fig. 4e). Two phenomena become evident: (1) relative compositions of C=C isomer proportions vary amongst different classes of GPLs, and the n-7, n-9, and n-10 isomers are the major components; (2) LPC, PC, and PI contain n-13 as an additional isomer component. These observations indicate variations in the unsaturation profile across different phospholipid classes, which would be overlooked by pooled FA analysis alone.

## Analysis of paired normal and cancerous human bladder tissue samples

Lipidomic profiling is increasingly used as a direct readout to uncover altered lipid metabolism in cancers[58]. The HILIC-TIMS-MS/MS system offers a distinct capability to probe dysregulated lipid metabolism at the isomer levels which is missing in traditional lipidomic profiling methods. In this work, human bladder cancer tissue is used for demonstration. SPLASH (100 μL) was added to 50 mg tissue before lipid extraction, which provided a dynamic range of 0.05-15 for relative quantitation ($I/I_{IS}$) of each lipid class. An RPLC-MS/MS protocol was used to profile TGs at the sum composition level, while the developed HILIC-TIMS-MS/MS system was employed to analyze phospholipids at the C=C location level. The data revealed higher levels of phospholipids and lower levels of TGs in the cancer tissue relative to those in the normal tissue (Fig. 5a and Supplementary Data 5). These observations are consistent with previous reports regarding the altered TG and GPL compositions in bladder cancer tissue[59–61].

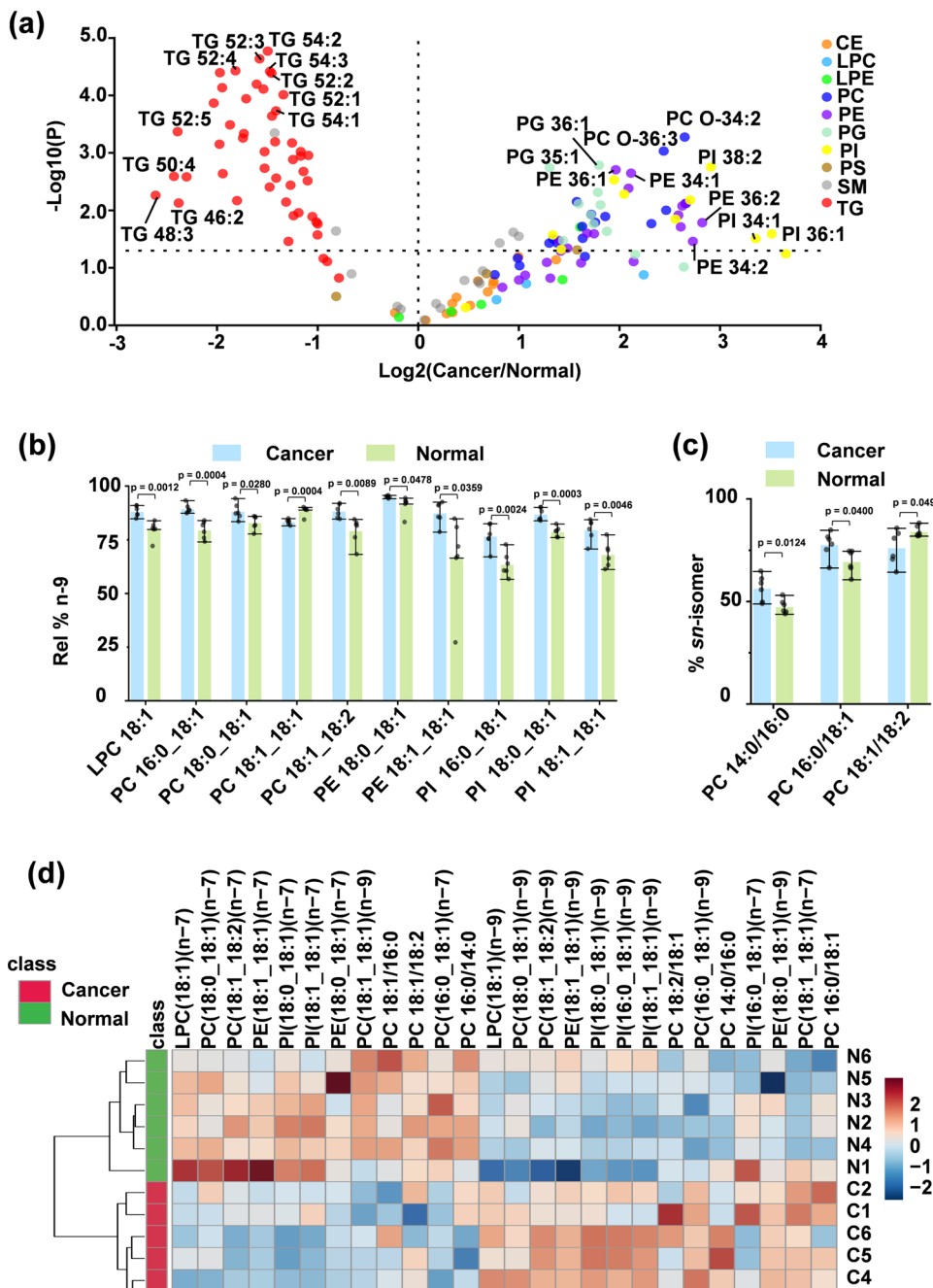

**Fig. 5 | Deep profiling of lipids in paired human bladder cancerous and adjacent normal tissues (N = 6).** Differences between the two groups of samples were evaluated for statistical significance using the two-tailed Student's *t* test. Error bar represents ±s.d. (N = 6). **a** Alterations of TG, CE, GPL, and SM at the sum composition level. SPLASH Lipidomix was used as the internal standard for relative quantitation. **b** Rel.% n-9 in C18:1 bearing GPLs, and **c** %*sn*-isomer in PC 14:0_16:0, PC 16:0_18:1, and PC 18:1_18:2. Bars represent mean values ± SD. **d** Hierarchical cluster analysis of paired human bladder cancerous (C1-C6) and adjacent normal tissues (N1-N6) using 20 GPL C = C location isomers and 6 PC *sn*-isomers. Colors represent relative amounts as indicated by the color bar. Source data are provided as a Source Data file.

We then explored whether the phospholipidome could be used for the phenotyping of cancerous vs. normal bladder tissue. Due to the large heterogeneity of bladder tissue samples, identified phospholipids at the sum composition level were in the range of 221 to 292 for the cancerous tissues (N = 6), while the numbers were in the range of 134 to 198 for the six paired normal tissues. Among these, 87 phospholipids were commonly detected and 69 showed significant changes (P < 0.05). However, hierarchical cluster analysis using the 69 phospholipids failed to discriminate the cancerous tissues from normal tissues (Supplementary Fig. 11).

Deep-profiling identified 95–181 GPLs at the C=C location level in the normal tissue samples while the numbers were increased to 155-334 in the cancerous tissue samples because the latter had higher contents of GPLs (Fig. 5a). We selected 15 commonly found C18:1–bearing GPLs for relative quantification. Due to the low abundances (<1%) of the n-10 isomer in GPLs, we only conducted relative quantitation of C18:1 n-9 in GPLs. The relative abundance of the n-9 isomer (rel.% n-9) was calculated by normalizing the ion abundance of n-9 C=C diagnostic ions to the total of n-9 and n-7 C=C diagnostic ions. 10 out of 15 C18:1–bearing GPLs were found to have significant changes

in rel.% n-9 ($P < 0.05$). Besides PC 18:1_18:1, rel.% n-9 in 9 isomer pairs all exhibited significant increases in bladder cancerous tissue than in the normal tissue (Fig. 5b). For instance, Rel.% n-9 in PI 18:0_18:1 is 87 ± 3% in bladder cancer relative to 79 ± 2% in the control ($P = 3.2 \times 10^{-4}$). We found that the individual-to-individual variation was large for relative quantitation at the sum composition level, with an average RSD of 62 ± 23%. As a comparison, the average RSD was decreased to 5 ± 6 % for rel.% n-9. Besides C = C location isomers, the compositions of PC *sn*-isomers were also found altered in cancerous tissue samples. Three pairs of PC *sn*-isomers, including PC 14:0_16:0, PC 16:0_18:1, and PC 18:1_18:2 showed statistically significant changes in isomeric compositions in the cancerous tissue vs. normal control (Fig. 5c). Hierarchical cluster analysis using the above 20 C = C location isomers of GPLs and 6 PC *sn*-isomers allowed successful discrimination of bladder cancerous tissues from normal tissues (Fig. 5d). Overall, this set of data provides strong experimental evidence that profiling at detailed structural levels is useful in lipidomic phenotyping.

## Discussion

In this work, we have developed a HILIC-TIMS-MS/MS system for profiling of phospholipidome at deep structural levels and with distinct analytical advantages of fast speed, high sensitivity, and wide coverage. As compared to traditional RPLC methods employing relatively long time for lipid separations, the HIILIC-TIMS configuration benefits from the two fast orthogonal separations, thus largely improves the analysis for isobaric lipids and low abundance lipids. Although *sn*- and C = C location isomers cannot be separated on HILIC-TIMS, the isomer-resolved MS/MS methods fill the gap for their identification and quantitation. Moreover, the CCS values extracted from TIMS are proved as useful descriptor, further enhancing the confidence for lipid identification. We believe such a system is one of the best so far which provides a comprehensive and easy-to-adopt solution to lipidomic profiling where isomer analysis becomes increasingly important and necessary for studying lipid metabolism and lipidomic phenotyping. In our studies, deep profiling of phospholipids in bovine liver showcases the powerful analytical capability of the developed system. More than 400 phospholipids have been identified down to the C=C location level, with relative compositions of many C=C location isomers and *sn*-isomers of PC being mapped. Regarding lipid coverage, this number doubles as compared to that identified by a HILIC-PB-MS/MS system before[36], mainly due to discovering more low abundance lipids. To our surprise, a multitude of fatty acyls with unusual site(s) of unsaturation is found from RAW 264.7 macrophages that are not described by canonical pathways. Their identities are confirmed by correlating the changes of total fatty acid profile to the changes in the desaturase activity, including SCD1 and FADS2. Consequently, more than one thousand distinct structures of phospholipids are identified from RAW 264.7 cells. This large structure diversity highlights the need to perform isomer analysis for lipidomic profiling. Quantitation of structurally resolved lipids also has implications for disease diagnosis. A wide variety of isomers show significant compositional changes in human bladder cancer tissues relative to normal control, while such changes would be masked without differentiating isomers. It should be noted that in order to maintain a relatively high MS/MS throughput, the resolving power of TIMS is compromised to be around 100 on the current system; therefore, a baseline separation cannot be achieved for several types of isobaric lipids. This limitation can be addressed by employing either high-resolution IMS[62–65] or high-resolution demultiplexing[66]. Overall, the developed system shows a relatively complete solution for deep profiling of lipids that can be applied to both fundamental and clinical studies.

## Methods

### Sample preparation

All lipid standards were purchased from Avanti Polar Lipids, Inc. (Alabaster, AL, USA). All organic solvents, salts, and PB reagents were purchased commercially (details provided in the Supplementary information). RAW 264.7 macrophages (American Type Culture Collection (ATCC); Manassas, VA, USA) were used. The human bladder cancer tissue samples, including one Asian female and five Asian males, were provided by Peking University First Hospital. All the procedures related to these samples were compliant with all relevant ethical regulations set by the Ethical Review Board of Tsinghua University (IRB No. 2017007). Informed consent was obtained from all participants. Details of cell culture and lipid extraction[67] are provided in the Supplementary Information. A biological sample-specific total fatty acid database was created by means of saponification, 2-acpy derivatization, and subsequent RPLC-MS/MS (details provided in the Supplementary Information). An RPLC-MS/MS protocol applied to profile TGs in total lipid extract is provided in also the Supplementary Information.

### PB derivatization

Lipid extract and 10 mM triFAP were dissolved in 200 μL ACN/MeOH/water (50/2/48, v/v/v). The solution was purged with nitrogen gas to remove residual oxygen before the PB reaction. The solution was injected into the flow microreactor for 20 s' UV irradiation (254 nm). The PB reaction solution was collected in a glass vial for subsequent HILIC-TIMS-MS/MS. For PG and PI, a modified TMSD methylation[68] was employed before triFAP derivatization(details in the Supplementary Information).

### HILIC-TIMS-MS/MS

A Waters ACQUITY UPLC I-Class system (Waters, Milford, MA, USA) was used to separate polar lipids on a CORTECS UPLC HILIC column (150 mm × 2.1 mm, 1.6 μm, Waters, Milford, MA, USA). Detailed HILIC separation is provided in Supplementary Information. HILIC-TIMS-MS/MS detection was performed using a Bruker timsTOF instrument, equipped with an ESI source. A detailed description of the mass spectrometer has been reported elsewhere[55]. The dual TIMS setup allows operating the system at 100% duty cycle when accumulation and ramp times are kept equal. Data acquisition was performed using otofControl (version 5.0, Bruker Daltonics, Bremen, Germany). Here, we set the accumulation and ramp time to 40 ms each and recorded mass spectra in the range from *m/z* 100–1500 in both positive and negative electrospray modes. Precursors for data-dependent acquisition were isolated within ±1 *m/z* and fragmented with *m/z*-dependent collision energy, which was linearly increased from 25 to 45 eV in positive mode, and from 35 to 55 eV in negative mode. The overall acquisition cycle of 0.2 s comprised one full TIMS-MS scan and four TIMS-MS/MS scans. Low-abundance precursor ions with an intensity above a threshold of 100 counts but below a target value of 4000 counts were repeatedly scheduled and otherwise dynamically excluded for 0.1 min. Tuning Mix (ESI-L, Agilent, Santa Clara, CA, USA) was used to perform calibration for *m/z* and ion-mobility values.

### Data analysis

The analysis of HILIC-TIMS-MS/MS data was carried out with a home-made software, LipidNovelist. Data Analysis 5.0 software (Bruker Daltonics, Bremen, Germany) was used to convert raw data to an open file format (.ascii), which can be read by LipidNovelist. For relative quantitation at the sum composition level, MS[1] spectra of specific lipid classes and corresponding internal standards were imported into the LipidNovelist Extension and subjected to Type-I isotope correction (Supplementary Note 2). LipidNovelist read the data from HILIC-TIMS-MS/MS in negative ion mode and provides lipid identification at the fatty acyl (alkyl) chain level. The lipid class search comprised PC, PC-O, PE, PE-O, PG, PI, PS, PnE, SM, LPC, and LPE. The precursor ion and MS/MS fragment mass tolerances were set to 10 ppm, respectively. Lipid chain level identification was carried out by class-specific retention

times in HILIC, accurate *m/z*, MS/MS fragments based on lipid class-specific fragmentation rules. LipidNovelist also conducted de novo analysis of the data generated from HILIC-TIMS-PB-MS/MS and the C = C location of each unsaturated acyl (alkyl) chain was assigned based on the detection of C = C diagnostic ion pairs. LipidNovelist can also calculate the abundance ratios of diagnostic ions corresponding to C = C location isomers for relative isomer quantitation.

## Reporting summary

Further information on research design is available in the Nature Portfolio Reporting Summary linked to this article.

## Data availability

All data necessary to support the conclusions are available in the manuscript or supplementary information. The raw MS data are available from Figshare (https://doi.org/10.6084/m9.figshare.21946100)[69]. Source data are provided with this paper.

## Code availability

The source code of LipidNovelist is freely available for academic users from the corresponding authors upon request. The most up-to-date version of LipidNovelist & LipidNovelist Extension, along with instructional videos and example data to facilitate its utilization, can be freely accessed at https://doi.org/10.6084/m9.figshare.22297771.

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

## Acknowledgements

Financial support from the following funding agencies is greatly appreciated: National Key R&D Program of China (2018YFA0800903), National Natural Science Foundation of China (No. 22225404, 21825702, and 22137004), Beijing Outstanding Young Scientist Program (BJJWZYJH01201910003013), and Beijing Advanced Innovation Center for Structural Biology Funding (20151551402).

## Author contributions

Y.X., T.X., H.Y., and Y.G. designed the experiments. T.X. performed the research; X.J and H.S. assisted in performing experiments, T.X., D.Z., F.Z., and Y.G. analyzed the data. T.X. and Y.X. co-wrote the paper. All authors discussed the results and commented on the paper.

## Competing interests

The authors declare no competing interest.
