## [Peer Review File · Nature Communications]

Deep-profiling of phospholipidome via rapid orthogonal separations and isomer-resolved mass spectrometryEditorial Note: Parts of this Peer Review File have been redacted as indicated to remove third-party material where no permission to publish could be obtained.

REVIEWER COMMENTS

Reviewer #1 (Remarks to the Author):

This manuscript reports on the use of a complex methodology for deep profiling of the phospholipidome. In general, the manuscript is well written, so it is easy to follow the workflows and the discussed results. The described approach combines several methods, and in total, it provides an impressive coverage of the lipidome. However, I would like to suggest some aspects to which attention should be paid.

The important issue of manuscripts published in Nature journals is the scientific novelty. The group of Yu Xia is known to be one of the leading groups in lipidomic analysis with special focus on detailed structural characterization. They published a series of high-quality and innovative papers in previous years showing several new methods. One may have the feeling that this manuscript combines several previous works into one more comprehensive with some improvements of individual methods and the added value in the combination of these approaches, which is certainly a valuable contribution, but from the point of view of novelty, many important parts are referenced to previous papers of this group, e.g., the HILIC method [45], the use of bicarbonate adducts [40, 45], the use of triFAP [34], the use of Patterno Buchi [33], the lipid annotation software [47]. However, I really miss one essential part, which is the quantification. I regret to write that I do not find anything related to the quantitation, such as the use of appropriate exogenous internal standards (preferably isotopic labelled), validation of the analytical method, suitable quality controls, etc. It has been shown many times that MS-based quantitation without the use of internal standards, the quantitation is not reliable, and it may be difficult to reproduce such data by other groups. I think that quantitation could be a challenge here because individual lipid species will be separated by ion mobility by MS/MS, therefore the correlation to IS could be complicated, but this aspect could add real scientific novelty, in line with what I would expect for this journal. If I understand well, this work shows only the relative ratios of peak intensities without any use of IS, which is also not consistent with the recommendation of the International Lipidomics Society.

Particular comments:

1/ I have enjoyed reading of Introduction except for one issue where I cannot agree. Authors emphasize the role of ion mobility, but underestimate the role of chromatography and, in general, the potential of chromatographic techniques for the separation of lipid isomers is not fairly reported. Various chromatographic modes can be used for the resolution of all types of lipid isomers that differ in the position of the double bond, geometry of the double bond, fatty acyl chain branching, and even enantiomers can be separated on the chiral column. For example, the authors show a partial ion mobility separation of PC 16:0/18:1 and PC 18:0/16:0 with relatively minor resolution, but this mixture can be easily resolved to the baseline using RP-LC. I fully respect the important role of ion mobility, but some changes in the Introduction and text should be modified to report it in a more balanced way.

2/ It is quite odd to cite only the work of Vasilopoulou et al. [14] with the claim of increasing coverage of lipidome, but omit the related Matters arising (Nat. Com. 12 (2021) 4771), which clearly illustrated a significant portion of misidentification in the work. This could confuse the readers because they may have the feeling that the list of reported lipids in [14] is correct. This should be given into the right consequences here.

3/ In the abstract and maybe elsewhere, the authors report on 5 min LC run for HILIC analysis. This could anticipate that they can do 12 injections per hour, which is certainly not true because the HILIC method requires longer equilibration times in the case of full equilibration or moderate time for partial equilibration. I suggest reporting also the total analysis time (including re-equilibration), which is best illustrated by the real number of injections per hour or per day.

4/ I would suggest adding one summary table of all identified lipid species in the supplementary part with their summary formula, retention times, and collision cross-sections. This information is probably shown in a few supplementary tables, but one summary table would be helpful for readers.

5/ I cannot find information on whether the authors use also Type I isotopic correction, which is

important for quantitation. It is relatively easy to apply.

6/ Lines 150-161 – authors report that the ion mobility separation with much slower scan rate (0.06 V/ms) is better, however the separation shown in Supplementary Figure 3 is not satisfactory. I would suggest showing the separation for both settings (1 V/ms and 0.06 V/ms) and also provide the information of the time needed for a single scan, which is shown only for the faster scan (40 ms).

7/ Fig. 2 – I suggest adding the time scale for isomer-resolving MS/MS, similarly as for other techniques. You mention 100 microseconds in line 215, but it is not clear whether this corresponds to MS/MS or full scan MS.

8/ The annotation of ether-phospholipids does not follow the recommended shorthand annotation published in the Journal of Lipid Research. The abbreviation “O” should be used for ethers and “P” for plasmalogens, but not “E”, as written, for example, in lines 260-261.

9/ Discussion on page 17 on cancer tissue samples – you analyze human bladder cancer tissue samples, but the first citation in this paragraph [51] is on colorectal cancer, which could differ. At least, this difference should be commented. Moreover, this is an unusual example in which TG levels in cancer tissues decrease and SM increase because some references and also our experience show the opposite. You should be more careful with the conclusion here. It would require stronger justification for such claims.

10/ Line 470 – the use of Thomson (Th) for m/z is discouraged by IUPAC recommendations for mass spectrometry. Please replace by m/z according to the recommendations.

Reviewer #2 (Remarks to the Author):

This manuscript describes the profiling of phospholipids in a range of contexts and aims to demonstrate enhanced structural elucidation of lipids to expand our knowledge of the composition of the lipidome. Critical aspects of the analytical pipeline have been showcased before by this group (including in Nat. Commun.). Specifically, the integration of HILIC chromatography with rapid ex situ derivatization followed by tandem mass spectrometry for identification of carbon-carbon double bond positions and thus discrimination between regioisomers. Similarly, the utilisation of methylation of some phospholipid classes and the use of carbonate buffers to access structure-sensitive fragmentation patterns have been well-described previously by this group and others. The new aspect here is in the integration of trapped ion mobility spectrometry into the workflow. The authors do a good job of rationalising and demonstrating how the additional dimension of separation assists in resolving isobaric complexity within lipid subclasses that are typically compressed into a single narrow chromatographic peak by HILIC. The exploration of the parameter space and the demonstration of the improvements for isobaric and isomeric resolution will be interesting to the specialist but overall the result does not appear to this reader as a single integrated method that provides significantly more information than previous works. For example, the workflows to obtain information on sn-position and double bond position appear to be separate pipelines and the data are not aligned or merged. Indeed in the latter part of the manuscript where the applications are described the bicarbonate analysis does not appear to have been undertaken (or at least the results are not provided). From an analytical point of view the new approach does not seem to be benchmarked against other work, for example, the RAW 264.7 cells have been exhaustively studied by the LipidMAPS consortium how do the results of these new analytical pipelines compare to existing knowledge of this lipidome? The application of the methods to cancer seems to yield some very interesting results but these appear to be important but confirmatory rather than novel and are somewhat lost in this paper which is so methodologically focussed. This aspect of the work would stand alone as an interesting contribution to lipid cancer biology. So overall, the analytical advance would be more suited to a specialist journal where the contribution of the mobility dimension to the analysis can be more fairly assessed, while the cancer lipidomics would be better promoted to a study in its own right. Other aspects for the authors consideration are provided below:

1. Title, pg 1. “isomer-resolving” should be “isomer-resolved”

2. Abstract, pg 2. Claim of nM sensitivity refers to on-column and not the sample. A more robust metric is needed throughout that would allow the reader to assess the claim against prior work. E.g., number of cells, volume biofluid, dry weight tissue etc. The work on the well studies RAW cell line

enables this.

3. Introduction, pg 3, line 50 “accompanied by isomers” should be “that includes isomers” the isomers are part of the lipidome not something that accompanies it.

4. Introduction, pg 3, line 59 IM-MS is a better acronym for ion mobility mass spectrometry

5. Introduction, pg 3, line 62 The claims made in reference 14 and recapitulated in this introduction have been strongly contested. The authors should review subsequent commentary to this manuscript and make its own critical evaluation of the claims rather than simply parroting them.

6. Introduction, pg 3, line 69 should be “in the studies above”

7. Introduction, pg 4, line 72 “relative geometry” should be “relative position” geometry refers to stereoisomers not regioisomers as being discussed here.

8. Introduction, pg 4, line 83 “abnormous” is an unusual word. One would normally use “abnormal”. In this context however it is rather subjective to say what is normal versus abnormal. Suffice to say that the lipid metabolism is “altered” in disease.

9. Results, pg 6, line 119, “differing” rather than “differ”

10. Results, pg 7. The discussion of the sn-positional isomers in the standard is somewhat confusing in its presentation with the purity percentage of a reference material variously measured as 66% and 82%. The authors should clarify what the purity of the reference material is (by an independent method, e.g., PLA2 enzyme) and then discuss how peak abundances vary depending on the mobility conditions. There is a broader unknown here if this approach is valid for other phospholipids? Other CID based (PD and OzID-based) methods all show some dependence on the acyl chain identity on the product ion abundance so it is unclear how universally this bicarbonate approach can be deployed without calibration by reference standards. Along these lines, one would expect that some sn-positional isomers will also show some mobility-based resolution; notably where acyl chains differ significantly in chain-length or unsaturation. It would be of interest therefore to examine these data sets for evidence of this.

11. Results, pg 9, line 194 and Fig. 2. The phosphate methylation for PG and PI lipid classes seems to be mentioned in passing and it is not shown at all in Fig. 2. It is unclear whether this is a crucial part of the workflow in order to analyse these classes or whether it represents an add-on for deeper profiling of these classes. Overall I was left wondering what the overall workflow looks like and how many parallel analyses (injections) are required for the final data set underpinning the title claims of sensitivity and lipidome coverage? This should be clearly defined.

12. Results, pg 9, line 200. This seems to me to be the major technical drawback of the method in that PB-derivatization of a monounsaturated lipid generates two isomers that then serves to broaden the mobility distribution. Imagine how this effect is confounded in a polyunsaturated lipid where each double bond generates two isomers that have slightly different collision cross sections. Take the extreme case of PE 38:6 this would lead to 12 isomers that could be further doubled to 24 in the presence of native sn-positional isomers. Decreases in resolution will lower limits of detection and confound resolution of native isomers. The authors need to clearly articulate this issue and provide enough examples to show whether this is a problem or not.

13. Results, pg 12 and throughout. Please use agreed nomenclature for PE ethers (Leibisch et al.) recommend PE O-35:3 where the identify of the ether linkage is unknown. Also in this section bovine liver has previously been subject to extensive investigation. One of the findings is that it has high levels of mono- and di-methylated PE. See doi:10.1016/j.bbaliip.2011.09.018 and DOI: 10.1021/acs.analchem.5b02243. Indeed, these isomers may help explain some of the more unusual odd chain polyunsaturates that are assigned in this section or at the very least they would not be ruled out by the data. For example, PE O-17:1/18:2 could be MMPE O-16:1/18:2 or PE 19:3_16:0 could be MMPE O-18:2/16:0. It would be interesting to see whether ion mobility can also be used to classify these subclasses of PE.

14. Results, pg 12. Similar to point #13 the putative “discovery” of phosphonate GPLS appears significant but then more evidence should be provided to support this. Is there a reference CID spectrum or reference standard that can be used?

15. Results, pg 15, line 316. “we were motivated” not “are”

16. Results, pg 15, line 328. “proposed” not “demonstrated”. In general the authors should be careful to articulate what are associations and what are demonstrations.

Reviewer #3 (Remarks to the Author):

This manuscript from Xia et al. reports on a comprehensive lipidomics strategy incorporating solution phase derivatization and bond-specific UV activation with liquid chromatography-ion mobility-tandem mass spectrometry (LC-IM-MS/MS) analysis. The LC and IM separation dimensions are selective to resolving different phospholipid classes, whereas the MS/MS analysis provides precursor/fragment ion information that is used to provide structural details for each isolated m/z . Specifically, the negative polarity MS/MS provides acyl chain length information and sn - stereolocation, while positive polarity MS/MS is combined with offline derivatization/bond activation to determine double bond location(s). The downstream data analysis is achieved using software developed in-house by the authors, which is necessary given that no commercial software can currently support this workflow. The authors demonstrate the profiling capabilities of this workflow on bovine liver lipid extract, RAW 264.7 cell line macrophages, and human bladder tissue. The liver extract was used to assess the analytical figures-of-merit of the approach and evaluate the front-end derivatization strategy. An optimized workflow incorporating triFAP derivatization and Paternò–Büchi photochemical reactions was subsequently used for profiling macrophage-like cells from the cancer cell line RAW 264.7 and treated with small molecule enzyme inhibitors, demonstrating lipid inhibition and relative changes in the unsaturation compositions. Of note was that over one thousand structurally-distinct lipids were annotated from the cell line, which is approximately 2-3x more than is typically reported in similar lipidomic studies. Finally, the deep-profiling workflow was applied to normal and cancerous human bladder cells, identifying stereo-specific lipid changes.

Overall, this is an impressive body of work and demonstration of an isomer-specific lipidomic workflow that, unlike other studies that claim deep lipidomic profiling, this work is clearly an in-depth profiling approach. The analytical findings are well-grounded by the data, and the amount of experimental detail provided is sufficient for others to reproduce this work. As such, this work is recommended for publication, after the authors consider some minor revisions, outlined below.

General Comments:

1. The authors demonstrate that the ion mobility can help offset some of the limited resolution of the chromatography stage while also improving signal abundance and the quality of mass-selected MS/MS experiments. A TOF mass spectrometer is used here, but the resolving power achieved is not stated. In figure 3, panel B, a partially-resolved mass spectrum is shown, but how much more mass resolving power would be needed to offset the need for the ion mobility dimension? That is to say, is high mass resolution enough (e.g., Orbitrap) to see similar deep-profiling results, or are other analytical merits needed, such as the accumulation afforded by TIMS or limitations associated with the quadrupole mass isolation window? Some of these points are recommended to be discussed in the manuscript.
2. The ion mobility spectra shown (Fig 1D, Fig 3C) are only partially-resolved, and the authors use a Gaussian deconvolution to determine relative abundances. This partial resolution brings up a question of: how important higher ion mobility resolution would be to these studies? The authors discuss a compromise between TIMS scanning speed and resolution, but then what about other high-resolution ion mobility, fast scanning capabilities now emerging (such as SLIM or deconvolution approaches)? Can the authors comment on the benefits that higher IM resolution would provide for these studies, or in their opinion, is this about as good as it will get in terms of lipidomic coverage?
3. Regarding the ion mobility, the authors discuss the inverse ion mobility $1/K$, but do not work with collision cross section, which many other groups are adopting for these studies to improve molecular identifications. Would cross sections or reproducible $1/K$ values be useful for this workflow? How reproducible is the inverse mobility, and can it be used as a lipid descriptor at the specific structural level?

4. Please ensure that the software (LipidNovelist) and databases used in this study are made freely-available. It does appear that the authors have provided these as part of their submission.

5. Some of the authors' findings clearly point to the impurity/unavailability of current commercial lipid standards (e.g., page 7, line 142; page 15, line 334). This is less of an issue for this work and more of an important issue that should be raised more often whenever possible. The comment here is one of appreciation that the authors are helping to bring the inadequacies of currently-available lipid standards to light.

Other minor issues of note:

(Page 3, line 66) "...by adding added..." can be revised to "...by including...".

(Page 6, line 133) "...separates of..." can be just "...separates...".

(Page 7, line 160) "...time needed of..." can be "...time needed for...".

(Page 9, line 205, line 207) The use of "C=Cs" is confusing. Recommend "C=C bonds" here.

(Page 15, line 332) "...as well as together...".

(Page 18, line 388) The use of "Rel% n-9" is somewhat confusing. It is repeated enough to warrant this abbreviation, but perhaps introduce it on first use, e.g., "The relative abundance of the n-9 isomer (rel% n-9)...".

(Page 20, line 432) "RAW"

Reviewer #4 (Remarks to the Author):

The manuscript by Xia et al described novel approach for deep profiling of lipids based on the PB derivatization, HILIC-TIMS-MS analysis. The overall methodology is very interesting. No doubt it is the next step in our efforts in exploration of natural lipidomes. Thus, I would strongly advise to publish the manuscript as it is important step in technology development (although some parts are already published by others as well) but after major revision.

My main concern is in the current form manuscript is rather poorly structured and hard to follow. It is not written in enough details for bioanalytically oriented readers, but also does not have real biological insights which can be important for more biomedical oriented audience. Authors needs to decide which audience they would like to target primary with this work.

Below are some specific points which might be useful for manuscript revision:

Unique point of methodology should be mentioned in the abstract; otherwise it is rather generic and unattractive .

General proof reading is needed – some sentences are not finished, e.g. p7, line 152, and others.

If the method is based on DDA (classical one, without inclusion list), how then particular charged forms (adducts or deprotonated ions) are specifically selected for MS/MS events?

100 uL of SPLASH for 50 mg of tissue seems really a lot to me... Why such high amount of internal standards was selected? Is it still in the dynamic range of the internal lipids which were quantified relative to the SPLASH standards?

How many cells were used for the experiments?

It is claimed that method take 5 min (even one HILIC run was actually 6 min), but in reality the workflow actually takes much longer. It starts with FA saponification, derivatization and RPLC-MS analysis (7 min) of free FA to generate sample specific fatty acid database. TMSD derivatization and separate of Me PLs adds another 8 min HILIC run, as well as RPLC-MS/MS for TG identification (10 min) when needed. Then also PB derivatization of total lipid extract before HILIC-MS/MS. And what about combination of pos and neg ion modes? Detailed scheme representing the whole workflow with sequence of steps and brief explanations is needed to support readability of the manuscript. Figure 2 does not reflect the real composition of the methodology and thus very misleading.

3 provided application examples create more confusions than clarity.... In each of them different lipids are reported at the different levels of structural annotations, results are not really discussed appropriately from biological or biomedical perspectives, so readers will be left with the feeling that methodology produce a lot of data but usability of these data remains mostly unclear. I would suggest to focus on one example (and if wanted, move the other 2 into supplementary information) but present it in much more details and clarity, starting with data acquisition strategy ending up with discussion of biological significance.

Point-by-point response

Reviewer #1:

1. ...The important issue of manuscripts published in Nature journals is the scientific novelty. One may have the feeling that this manuscript combines several previous works into one more comprehensive with some improvements of individual methods and the added value in the combination of these approaches, which is certainly a valuable contribution, but from the point of view of novelty, many important parts are referenced to previous papers of this group, e.g., the HILIC method [45], the use of bicarbonate adducts [40, 45], the use of triFAP [34], the use of Patterno Buchi [33], the lipid annotation software [47].

Response: The reviewer is concerned about the novelty. We believe the developed system represents a new paradigm for conducting deep lipidomic profiling and the novelty resides in the following three aspects.

Firstly, this system harnesses the orthogonal separation capabilities of HILIC and TIMS into one system. As compared to the more commonly employed RPLC-IMS-MS configuration which relies on longer LC time (20 min to hours), our system achieves sensitive and wide coverage of phospholipids with fast separation (<10 min per LC run).

Secondly, because of better separation capabilities brought by HILIC-TIMS and the use of CCS as an additional descriptor, we remove the positive mode LC-MS/MS run which was necessary for class identification via the HILIC-PB-MS/MS system (2019 Nat. Commun.). Our new workflow directly employs DDA in the negative ion mode to acquire MS/MS spectra (>5000) in one LC run. This enables profiling at much higher throughput of phospholipids at the chain composition level and the discovery of a new class of phospholipid (PnE) from bovine liver lipid extract. These new capabilities could not be achieved via the previously reported methods.

Thirdly, we have developed a new method to create sample-specific FA database (C=C level) by correlating desaturase inhibition to the change of C=C location isomers in total FAs. The use of sample-specific FA database greatly improves the confidence for phospholipid annotation by LipidNovelist given that most standards for lipid C=C isomers are not commercially available.

Overall, the HILIC-TIMS-MS/MS system is the first platform that can provide deep profiling (C=C location, PC *sn*-position) of phospholipidomes at high sensitivity (nM) and with wide coverage (>400 lipid detailed structures). It is proved to be powerful in discovering new lipid classes (e.g., PnE), uncanonical desaturation pathways (RAW 264.7 cell line), and phenotyping disease tissue samples (human bladder cancer).

Change: We have revised the Abstract and discussions in the main text (page 13) to emphasize the logic behind the designed experiments and associated novelty.

2. However, I really miss one essential part, which is the quantification. I regret to write that I do not find anything related to the quantitation, such as the use of appropriate exogenous internal standards (preferably isotopic labelled), validation of the analytical method, suitable quality controls, etc. It has been shown many times that MS-based quantitation without the use of internal standards, the quantitation is not reliable, and it may be difficult to reproduce such data by other groups. I think that quantitation could be a challenge here because individual lipid species will be separated by ion mobility by MS/MS, therefore the correlation to IS could be complicated, but this aspect could add real scientific novelty, in line with what I would expect for this journal. If I understand well, this work shows only the relative ratios of peak intensities without any use of IS, which is also not consistent with the recommendation of the International Lipidomics Society.

Response: We performed relative quantitation of phospholipids on three structural levels. SPLASH was used as the internal standard mixture for relative quantitation of GPLs at the sum composition level. Because lipid standards are very limited at C=C location and *sn*-levels, relative quantitation of isomers was based on detecting the diagnostic fragment ions in the same MS/MS spectrum (no internal standard used).

Change: To clarify that relative quantitation was performed with the use of SPLASH, the following sentences are added.

Page 20, “ ...SPLASH (100 μ L) was added to 50 mg tissue before lipid extraction, which provided a dynamic range of 0.05-15 for relative quantitation (I/I_{IS}) of each lipid class.”

Page 26, Methods:

“For relative quantitation at the sum composition level, MS¹ spectra of specific lipid classes and corresponding internal standards were imported into the LipidNovelist Extension and subjected to Type-I isotope correction (Supplementary Note 2).”

Particular comments:

1. I have enjoyed reading of Introduction except for one issue where I cannot agree. Authors emphasize the role of ion mobility, but underestimate the role of chromatography and, in general, the potential of chromatographic techniques for the separation of lipid isomers is not fairly reported. Various chromatographic modes can be used for the resolution of all types of lipid isomers that differ in the position of the double bond, geometry of the double bond, fatty acyl chain branching, and even enantiomers can be separated on the chiral column. For example, the authors show a partial ion mobility separation of PC 16:0/18:1 and PC 18:0/16:0 with relatively minor resolution, but this mixture can be easily resolved to the baseline using RP-LC. I fully respect the important role of ion mobility, but some changes in the Introduction and text should be modified to report it in a more balanced way.

Response: We agree with the reviewer that comprehensive LC methods have been reported for the separation of lipid isomers but they typically require longer separation time (20 min to hours).

This is exact the motivation for us to couple fast LC separation (HILIC) with ion mobility and isomer-resolved MS/MS to achieve deep profiling in short time (< 10 min per run).

Change: Page 3, Introduction, the following sentences and associated references are added to emphasize the importance of LC for lipid analysis:

“The use of LC effectively mitigates ion suppression and isobaric interferences encountered in direct infusion methods. Additionally, dedicated LC methods have been developed to separate lipid isomers with variations in carbon-carbon double bond (C=C) position and geometry,⁹ as well as lipid phosphate regio-isomers^{10, 11} and enantiomers¹², albeit requiring moderate to long separation time (20 minutes to several hours).”

2. It is quite odd to cite only the work of Vasilopoulou et al. [14] with the claim of increasing coverage of lipidome, but omit the related Matters arising (Nat. Com. 12 (2021) 4771), which clearly illustrated a significant portion of misidentification in the work. This could confuse the readers because they may have the feeling that the list of reported lipids in [14] is correct. This should be given into the right consequences here.

Response/Change: Thanks for pointing this out. We changed the reference to a more recent publication by Lerner et al. (Nat. Commun. 2023, 937) and added comments on the importance of using retention time and CCS values to improve the accuracy of lipid annotation (Kofeler et al., Nat. Commun., 2021, 4771).

Page 3 and 4: The following sentences are added:

“Lerner et al. demonstrated high-throughput profiling of lipids on an MS platform consisting of microflow reversed-phase liquid chromatography (RPLC), trapped ion mobility spectrometry (TIMS), and parallel accumulation-serial fragmentation¹⁸. The utilization of four-dimensional data, including *m/z*, tandem mass spectra, LC retention time, and CCS values, enables the application of stringent criteria for lipid annotation, consequently reducing the extent of misidentifications, a crucial aspect that has gained considerable attention within the lipidomics community¹⁹”.

3. In the abstract and maybe elsewhere, the authors report on 5 min LC run for HILIC analysis. This could anticipate that they can do 12 injections per hour, which is certainly not true because the HILIC method requires longer equilibration times in the case of full equilibration or moderate time for partial equilibration. I suggest reporting also the total analysis time (including re-equilibration), which is best illustrated by the real number of injections per hour or per day.

Response/Change: We have changed 5 min to less than 10 min the Abstract and main text. The following description is also added.

On page 11, The following sentences are added:

“As shown in Fig. 2a, the first step involves data dependent analysis (DDA) in negative ion mode via a 6.5 min HILIC-TIMS-MS run (5 min separation and 1.5 min re-equilibration).”

4. I would suggest adding one summary table of all identified lipid species in the supplementary part with their summary formula, retention times, and collision cross-sections. This information is probably shown in a few supplementary tables, but one summary table would be helpful for readers.

Response/Change: Thanks for the suggestion. We have added the chemical formula, m/z , retention time, CCS into the supplementary tables. These tables are collated into an xlsx document.

5. I cannot find information on whether the authors use also Type I isotopic correction, which is important for quantitation. It is relatively easy to apply.

Response/Change: Thanks for the suggestion. We have added type I isotope correction into LipidNovelist Extension for relative quantitation at the sum composition level.

Page 11, the following sentence is added:

“For the acquired MS¹ data, LipidNovelist Extension performs relative quantification (I/I_s) at the sum composition level after Type-I isotope correction.”

Page 26:

“...For relative quantitation at the sum composition level, MS¹ spectra of specific lipid classes and corresponding internal standards were imported into the LipidNovelist Extension and subjected to Type-I isotope correction (Supplementary Note 2).”

6. Lines 150-161 – authors report that the ion mobility separation with much slower scan rate (0.06 V/ms) is better, however the separation shown in Supplementary Figure 3 is not satisfactory. I would suggest showing the separation for both settings (1 V/ms and 0.06 V/ms) and also provide the information of the time needed for a single scan, which is shown only for the faster scan (40 ms).

Response/Change: Thanks for the suggestion. We have added separation using 0.06 V/ms in Figure 1d, which corresponds to a scan time of 360 ms and an R of 195.

Page 8: The revised sentences are as follows:

“...Although better separation of the isobars can be achieved by using a scan rate of 0.06 V/ms ($R \sim 195$, Fig. 1d lower panel), it takes much longer scan time (360 ms vs. 40 ms), thus reducing the throughput of MS/MS analysis.

7. Fig. 2 – I suggest adding the time scale for isomer-resolving MS/MS, similarly as for other techniques. **You mention 100 microseconds in line 215, but it is not clear whether this corresponds to MS/MS or full scan MS.**

Response/Change: Page 10, the following sentence is revised to:

“ ... the acquisition speed of TOF for MS¹ and MS/MS is about 100 μ s per spectrum”

8. The annotation of ether-phospholipids does not follow the recommended shorthand annotation published in the Journal of Lipid Research. The abbreviation “O” should be used for ethers and “P” for plasmalogens, but not “E”, as written, for example, in lines 260-261.

Response/Change: We have changed PE-E to PE-O in the main text and supplementary information.

9. Discussion on page 17 on cancer tissue samples – you analyze human bladder cancer tissue samples, but the first citation in this paragraph [51] is on colorectal cancer, which could differ. At least, this difference should be commented. Moreover, this is an unusual example in which TG levels in cancer tissues decrease and SM increase because some references and also our experience show the opposite. You should be more careful with the conclusion here. It would require stronger justification for such claims.

Response/Change: Thanks for the suggestion. The discussion is revised to be specific to bladder cancer tissue. The same phenomenon (increased GPL and decreased TG) was reported by other groups and the following papers are cited:

- Tripathi P., et al., HR-MAS NMR Tissue Metabolomic Signatures Cross-validated by Mass Spectrometry Distinguish Bladder Cancer from Benign Disease. *J. Proteome Res.* 12, 3519-3528 (2013).
- Dill A.L., et al. Multivariate statistical identification of human bladder carcinomas using ambient ionization imaging mass spectrometry. *Chem. Eur. J.* 17, 2897-2902 (2011).
- Dobrzyńska I. et al., Characterization of human bladder cell membrane during cancer transformation. *J. Membr. Biol.* 248, 301-307 (2015).

Page 20: The sentences are revised as follows:

“Lipidomic profiling is increasingly used as a direct readout to uncover altered lipid metabolism in cancers⁵⁷. The HILIC-TIMS-MS/MS system offers a distinct capability to probe dysregulated lipid metabolism at the isomer levels which is missing in traditional lipidomic profiling methods. In this work, human bladder cancer tissue is used for demonstration. SPLASH (100 μ L) was added to 50 mg tissue before lipid extraction, which provided a dynamic range of 0.05-15 for relative quantitation (I/I_S) of each lipid class.”

“These observations are consistent with previous reports regarding the altered TG and GPL compositions in bladder cancer tissue⁵⁸⁻⁶⁰.”

10. Line 470 – the use of Thomson (Th) for m/z is discouraged by IUPAC recommendations for mass spectrometry. Please replace by m/z according to the recommendations.

Response/Change: Revision is made as suggested.

Reviewer #2:

1. ...The new aspect here is in the integration of trapped ion mobility spectrometry into the workflow. ... but overall the result does not appear to this reader as a single integrated method that provides significantly more information than previous works. For example, the workflows to obtain information on sn-position and double bond position appear to be separate pipelines and the data are not aligned or merged.

Response: The *sn*-position and C=C location are distinct structural levels for lipid identification and they typically require different MS/MS methods. *sn*-Specific C=C location information has been demonstrated by our group (Chem. Sci. 2019, 10, 10740) and others (Nat. Commun., 2020, 11, 375; J. Am. Chem. Soc. 2017, 139, 15681; Anal. Chem. 2016, 88, 2685). These methods all require multiple-stage ($n \geq 3$) MS/MS and result in lower sensitivity and throughput than reported in this work. Therefore, we did not pursue merging *sn*- and C=C for each identified phospholipid.

Change: Page 13, the following sentences are added:

“...Although the C=C location can be determined in phospholipids with *sn*-specific information^{21, 22, 34}, it requires multiple-stage MSⁿ ($n \geq 3$) which results in lower sensitivity and throughput. Therefore, we did not pursue merging the assignment of *sn*- and C=C location for each identified phospholipid in this workflow.”

2. Indeed in the latter part of the manuscript where the applications are described the bicarbonate analysis does not appear to have been undertaken (or at least the results are not provided).

Response: We would like to clarify that relative quantitation of both *sn*-isomers (PC) and C=C-isomers was performed and demonstrated to be useful for phenotyping cancerous human bladder tissue and normal tissue. The results are shown in Figure 5c,d and discussed in related text.

3. From an analytical point of view the new approach does not seem to be benchmarked against other work, for example, the RAW 264.7 cells have been exhaustively studied by the LipidMAPS consortium how do the results of these new analytical pipelines compare to existing knowledge of this lipidome?

Response: The LIPIDMAPS consortium organized quantitative analysis of the lipidome of RAW264.7 cell line on the sum composition level (Meth. Enzymol. 2007, 432, 21-57, J. Bio. Chem. 2010, 285, 39976). Our work is the first-time report on the unexpected diversity of C=C location isomers of phospholipids in RAW264.7 cells. The data are valuable resource to the lipidomics community given that RAW264.7 cell line is a frequently used model system in biological studies.

Change: On page 17 and 18, the following sentence are added:

“RAW 264.7 cell line is a frequently used model system for *in vitro* studies of immune responses. Notably, quantitative lipidomic profiling conducted by the LIPID MAPS consortium has facilitated a thorough investigation of the composition, biosynthesis, and function of the major lipid classes present in RAW 264.7 cells^{55,56}. Nevertheless, these studies mainly focused on lipid identification

and quantitation on the sum composition level. Herein, we aim to delineate the phospholipidome of RAW 264.7 cell line at the C=C location level, which has not been documented before and can serve as a valuable resource for studying the metabolism of unsaturated lipids.”

4. The application of the methods to cancer seems to yield some very interesting results but these appear to be important but confirmatory rather than novel and are somewhat lost in this paper which is so methodologically focussed. This aspect of the work would stand alone as an interesting contribution to lipid cancer biology. So overall, the analytical advance would be more suited to a specialist journal where the contribution of the mobility dimension to the analysis can be more fairly assessed, while the cancer lipidomics would be better promoted to a study in its own right.

Response/Change: We believe it is necessary to use clinical samples as a demonstration because clinical lipidomics will likely benefit most from the system developed in this study. We emphasize this point by showing that isomer quantitation (*sn*-for PC and C=C for GPLs) is much more effective in phenotyping the diseased and normal tissue with a small sampling size than traditional lipid profiling on the sum composition level. We have added discussions in the main text (page 13) to emphasize the logic behind the designed experiments and associated novelty.

5. Title, pg 1. “isomer-resolving” should be “isomer-resolved”.

Response/Change: Thanks for the suggestion. Revision is made throughout the manuscript and figures.

6. Abstract, pg 2. Claim of nM sensitivity refers to on-column and not the sample. A more robust metric is needed throughout that would allow the reader to assess the claim against prior work. E.g., number of cells, volume biofluid, dry weight tissue etc. The work on the well studies RAW cell line enables this.

Response/Change: The limit of detection of the analytical system was obtained via serial dilution. Therefore, the nmol/L sensitivity pertains to the sample instead of the on-column sensitivity. We added the injection amounts for each LC run, which included 20,000 cells and 100 µg bladder tissue per injection.

Page 10, the following sentence is revised to:

“...The sensitivity of was evaluated by serial dilution, with on-sample LODs achieved at 10 nM for PC 16:0/18:1(n-9) and PS 16:0/18:1(n-9),..”

Page 13, the following sentence are added:

“... the polar lipid extract of bovine liver (1 µg lipid extracts per injection)..... The performance of the system was further evaluated for profiling unknown phospholipidomes using RAW 264.7 cells(~20000 cells per injection)..... Finally, to demonstrate the unique capability of deep profiling in clinical applications, the human bladder tissue samples were analyzed (100 µg bladder tissue per injection).”

7. Introduction, pg 3, line 50 “accompanied by isomers” should be “that includes isomers” the isomers are part of the lipidome not something that accompanies it.

Response/Change: Thanks for the suggestion. Revision is made accordingly.

8. Introduction, pg 3, line 59 IM-MS is a better acronym for ion mobility mass spectrometry

Response: According to the literature, the full name for “ion mobility” should be “ion mobility spectrometry” and thus it is proper to use IMS as an abbreviation (ref: *Ion Mobility Spectrometry: Fundamental Concepts, Instrumentation, Applications, and the Road Ahead*. Dodds et al. JASMS, 2019, 30, 2185). Nowadays, ion mobility spectrometry hyphenated with mass spectrometry can be abbreviated as IM-MS. We think it is also better to use IMS in our context because the ion mobility module used in our system is *trapped ion mobility spectrometry* (TIMS).

9. Introduction, pg 3, line 62 The claims made in reference 14 and recapitulated in this introduction have been strongly contested. The authors should review subsequent commentary to this manuscript and make its own critical evaluation of the claims rather than simply parroting them.

Response/Change: Thanks for pointing this out. We changed the reference to a more recent publication by Lerner et al. (Nat. Commun. 2023, 937) and added comments on the importance of using retention time and CCS values to improve the accuracy of lipid annotation (Kofeler et al., Nat. Commun., 2021, 4771).

10. Introduction, pg 3, line 69 should be “in the studies above”

Change: Revised as suggested.

11. Introduction, pg 4, line 72 “relative geometry” should be “relative position” geometry refers to stereoisomers not regioisomers as being discussed here.

Change: Revised as suggested.

12. Introduction, pg 4, line 83 “abnormous” is an unusual word. One would normally use “abnormal”. In this context however it is rather subjective to say what is normal versus abnormal. Suffice to say that the lipid metabolism is “altered” in disease.

Change: Revised as suggested.

13. Results, pg 6, line 119, “differing” rather than “differ”

Change: Revised as suggested.

14. Results, pg 7. The discussion of the sn-positional isomers in the standard is somewhat confusing in its presentation with the purity percentage of a reference material variously measured as 66% and 82%. The authors should clarify what the purity of the reference material is (by an independent method, e.g., PLA2 enzyme) and then discuss how peak abundances vary depending on the mobility conditions. There is a broader unknown here if this approach is valid for other

phospholipids? Other CID based (PD and OzID-based) methods all show some dependence on the acyl chain identity on the product ion abundance so it is unclear how universally this bicarbonate approach can be deployed without calibration by reference standards.

Response: The bicarbonate method is only applicable to phosphatidylcholine (PC) because the gas-phase dissociation pathway is specific to the phosphocholine headgroup. Only *sn*-1 fragment ions are formed while the ions involving the *sn*-2 chain are absent. Consequently, there is no dependence of acyl chain identity which is a big advantage as compared to the traditional CID methods. Quantitation of the *sn*-isomers of PC via MS² CID of [PC+HCO₃]⁻ has been cross-validated by the PLA2 enzymatic approach with very small systematic deviation observed (2%) (Chem. Sci. 10, 2019,10740).

Change: We have revised the discussion regarding quantitation of *sn*-isomers on page 7.

“...Previously, we have established a quantitative analysis method for the *sn*-isomers of PC by detecting the *sn*-1 position specific ions, termed as “*sn*-1 fragment”, formed in MS² CID of [M+HCO₃]⁻ ⁴³. For instance, MS² CID of pure PC 18:0/16:0 only produces *sn*-1 18:0 ion (*m/z* 447.291, structure shown in the inset of Fig. 1), while that related to *sn*-2 16:0 should not be formed.”

Along these lines, one would expect that some *sn*-positional isomers will also shown some mobility-based resolution; notably where acyl chains differ significantly in chain-length or unsaturation. It would be of interest therefore to examine these data sets for evidence of this.

Response: The *sn*-isomers of most phospholipids share very similar CCS values (see Table below, values taken from the LIPIDCCS database, Anal. Chem. 2017, 89, 17, 9559). It was reported that super high resolving power (>500) of ion mobility could separate some *sn*-isomers (Anal. Chem., 2019, 91, 5021). We checked our data and did not observe any separation of the *sn*-isomers on our instrument.

	CCS A		
	[M+H] ⁺	[M+Na] ⁺	[M+HCOO] ⁻
PC 16:0/18:0	291.0	292.4	287.7
PC 18:0/16:0	291.0	292.4	287.7
PC 16:0/18:1	288.1	289.6	286.4
PC 18:1/16:0	288.1	289.6	286.4
PC 16:0/20:4	289.4	290.9	288.5
PC 20:4/16:0	289.4	290.9	288.5

Change: Page 8, the following sentences are added:

“It should be noted that the CCS values of the *sn*-isomers⁵⁰ of most phospholipids are very close to each other and it requires a resolving power (>500)⁵¹ much higher than that can be achieved on the current TIMS module. This again highlights the importance of using isomer-resolved MS/MS to achieve quantitation especially when the isomers cannot be separated.”

15. Results, pg 9, line 194 and Fig. 2. The phosphate methylation for PG and PI lipid classes seems to be mentioned in passing and it is not shown at all in Fig. 2. It is unclear whether this is a crucial part of the workflow in order to analyse these classes or whether it represents an add-on for deeper profiling of these classes. Overall I was left wondering what the overall workflow looks like and how many parallel analyses (injections) are required for the final data set underpinning the title claims of sensitivity and lipidome coverage? This should be clearly defined.

Response/Change: We have re-drawn Figure 2 to reflect all procedures. The workflow includes one -ve HILIC-TIMS-MS/MS for profiling at the sum composition and chain levels, two +ve PB + HILIC-TIMS-MS/MS runs for profiling at the C=C location level, and one +ve RPLC-MS/MS for establishing a sample-specific TFA database. Note that the RPLC-MS/MS run for TFA only needs to be conducted once for a specific type of sample. Discussion related to Figure 2 has been revised accordingly (page 8).

16. Results, pg 9, line 200. This seems to me to be the major technical drawback of the method in that PB-derivatization of a monounsaturated lipid generates two isomers that then serves to broaden the mobility distribution. Imagine how this effect is confounded in a polyunsaturated lipid where each double bond generates two isomers that have slightly different collision cross sections. Take the extreme case of PE 38:6 this would lead to 12 isomers that could be further doubled to 24 in the presence of native sn-positional isomers. Decreases in resolution will lower limits of detection and confound resolution of native isomers. The authors need to clearly articulate this issue and provide enough examples to show whether this is a problem or not.

Response: The reviewer has raised a valid concern. We found that the PB derivatization widens the peak width in mobility by 1.6 times, regardless of the number of C=C bonds present (Supplementary Table 1). However, all isomers resulted from the PB derivatization eluted as a single peak by TIMS and thus did not affect LOD in an obvious fashion. The data are provided in Supplementary S3 (^{PB}PE 38:3 and ^{PB}PE 38:4).

Change: Page 10, the following sentence is revised to

“...widened about 1.6-folds as compared to the remaining PE in the mobilogram. This phenomenon is consistently observed for polyunsaturated lipids such as PE 38:3 and PE 38:4. (Supplementary Table 1).

Supplementary Table 1. Comparison of the changes in full width at half maximum (FWHM) of the mobility peak of lipids before and after TriFAP PB derivatization.

	FWHM 1/k ₀ [M+H] ⁺	FWHM 1/k ₀ [M+triFAP+H] ⁺
PE 34:1	0.022	0.036
PE 36:1	0.023	0.038
PE 36:2	0.022	0.036
PE 38:3	0.022	0.036
PE 36:2	0.021	0.035

17. Results, pg 12 and throughout. Please use agreed nomenclature for PE ethers (Leibisch et al.) recommend PE O-35:3 where the identify of the ether linkage is unknown.

Response/Change: Thanks for pointing this out. We have changed PE-E to PE-O.

18. Also in this section bovine liver has previously been subject to extensive investigation. One of the findings is that it has high levels of mono- and di-methylated PE. See doi:10.1016/j.bbalip.2011.09.018 and DOI: 10.1021/acs.analchem.5b02243. Indeed, these isomers may help explain some of the more unusual odd chain polyunsaturates that are assigned in this section or at the very least they would not be ruled out by the data. For example, PE O-17:1/18:2 could be MMPE O-16:1/18:2 or PE 19:3_16:0 could be MMPE O-18:2/16:0. It would be interesting to see whether ion mobility can also be used to classify these subclasses of PE.

Response: The reference (doi:10.1016/j.bbalip.2011.09.018) conducted analysis on mouse liver while the McLuckey group (DOI: 10.1021/acs.analchem.5b02243) did not find any MMPE in the bovine liver lipid extract. MS² CID can readily differentiate the PE head group methylation vs. chain length difference. However, we did not identify any MMPE from +/- MS² CID from the same sample in this study or in our previous studies (2019, Nat. Commun.).

Change: Thanks for the suggestion on using ion mobility data to enhance classification of different phospholipids. We have added the following panels as Fig. 3i-k.

19. Results, pg 12. Similar to point #13 the putative “discovery” of phosphonate GPLS appears significant but then more evidence should be provided to support this. Is there a reference CID spectrum or reference standard that can be used?

Response/Change: Thanks for the suggestion. Unfortunately, we could not obtain synthetic standards for PnE from commercial sources. We have added the ion mobility data to show it is a distinct lipid class (Fig. 3k). Associated discussion is also added.

Page 16, the following sentences are added:

“It should be emphasized that the identified PnEs form distinct trend line in the CCS vs. m/z plot (Fig. 3j, k). PnEs are on average 0.5 Å² smaller than the PEs sharing the same acyl chain composition while 1 Å² larger than the corresponding isomeric PE-O (Fig. 3k). These findings further support that PnE is a distinct subclass of phospholipids and showcase that CCS values can be used as a descriptor to enhance lipid identification.”

20. Results, pg 15, line 316. “we were motivated” not “are”

Change: Corrected as suggested

21. Results, pg 15, line 328. “proposed” not “demonstrated”. In general the authors should be careful to articulate what are associations and what are demonstrations.

Change: Corrected as suggested.

Reviewer #3

“... The analytical findings are well-grounded by the data, and the amount of experimental detail provided is sufficient for others to reproduce this work. As such, this work is recommended for publication, after the authors consider some minor revisions, outlined below.

1. The authors demonstrate that the ion mobility can help offset some of the limited resolution of the chromatography stage while also improving signal abundance and the quality of mass-selected MS/MS experiments. A TOF mass spectrometer is used here, but the resolving power achieved is not stated. In figure 3, panel B, a partially-resolved mass spectrum is shown, but how much more mass resolving power would be needed to offset the need for the ion mobility dimension? That is to say, is high mass resolution enough (e.g., Orbitrap) to see similar deep-profiling results, or are other analytical merits needed, such as the accumulation afforded by TIMS or limitations associated with the quadrupole mass isolation window? Some of these points are recommended to be discussed in the manuscript.

Response: The TOF mass spectrometer employed in our study possesses a resolving power of 30,000. To fully resolve isobaric lipids such as PE 34:3 and PE O-35:3 ($\Delta_{\text{mass}} = 0.03639$ Da), it requires a resolving power exceeding 46,000 for based line separation. Although this resolving power can be satisfied by orbitrap, the bottleneck however is set by the resolving power of the quadrupole filter (~1000) for precursor isolation instead of product ion analysis. To confidently assign chain compositions, the two precursor ions need to be isolated to provide clean MS² CID spectra, which cannot be met by the quadrupole filter. Without TIMS separation, PE O-35:3 would be incorrectly identified as a mixture of PE O-17:0/18:3, PE O-17:1/18:2, PE O-19:2/16:1, and PE O-19:3/16:0, due to the detection of fatty acyl anions C16:1 (m/z 253.21), C16:0 (m/z 255.23), C18:3 (m/z 277.21), and C18:2 (m/z 279.23).

Change: Page 14, the following sentences are revised.

“As an example, PE 34:3 (m/z 712.4923) and PE O-35:3 (m/z 712.5287) can be partially resolved by the TOF mass analyzer which possesses a resolving power around 30,000 (Fig. 3b). Nevertheless, limitations associated with the isolation window of a quadrupole mass filter result in the co-isolation of lipid isobars during MS/MS experiment, producing a mixed MS/MS spectrum. PE(O-35:3) thus could be falsely identified as a combination of PE O-17:0/18:3, PE O-17:1/18:2, PE O-19:2/16:1, and PE O-19:3/16:0 due to the detection of fatty acyl anions C16:1 (m/z 253.25), C16:0 (m/z 255.23), C18:3 (m/z 277.21), and C18:2 (m/z 279.23) (Supplementary Fig. 9).

2. The ion mobility spectra shown (Fig 1D, Fig 3C) are only partially-resolved, and the authors use a Gaussian deconvolution to determine relative abundances. This partial resolution brings up a question of: how important higher ion mobility resolution would be to these studies? The authors discuss a compromise between TIMS scanning speed and resolution, but then what about other high-resolution ion mobility, fast scanning capabilities now emerging (such as SLIM or

deconvolution approaches)? Can the authors comment on the benefits that higher IM resolution would provide for these studies, or in their opinion, is this about as good as it will get in terms of lipidomic coverage?

Response: The resolving power of IMS needs to be higher than 400 for resolving the type II isobars and higher than 150 for separating isobars such as [PE 34:3-H]⁻ and [PE O-35:3 -H]⁻. Therefore, using other types of high-resolution or deconvolution IM methods will further improve the performance for deep lipidomic profiling.

Change: In Discussion, the following sentences are added.

It should be noted that in order to maintain relatively high MS/MS throughput, the resolving power of TIMS is compromised to be around 100 on the current system; therefore, a baseline separation cannot be achieved for several types of isobaric lipids. This limitation can be addressed by employing either high resolution IMS⁶¹⁻⁶⁴ or high-resolution demultiplexing⁶⁵.

3. Regarding the ion mobility, the authors discuss the inverse ion mobility 1/K, but do not work with collision cross section, which many other groups are adopting for these studies to improve molecular identifications. Would cross sections or reproducible 1/K values be useful for this workflow? How reproducible is the inverse mobility, and can it be used as a lipid descriptor at the specific structural level?

Response: Thanks for the suggestion and we have added CCS values to support lipid identification. We followed the standard calibration procedure and the 1/K₀ measured by TIMS was converted to CCS using the Mason-Schamp equation with an average CV less than 0.5% for intra-day repeatability. The CCS values obtained on several major lipids are comparable to other reports (Supplementary Table 2; Nat. Commun. 2023, 14, 937; Nat. Commun. 2019, 10, 985).

Change: In Supplementary Data 1, we document the measured 1/K₀ and CCS values of 228 lipid species at the sum composition level from bovine liver in negative ion mode. Additionally, scattered plots of *m/z* and CCS values of the above data are added as Fig 3i-k.

Supplementary Table 2. Comparison of experimental collision cross-section (CCS) values obtained in this work with those reported in the existing literature

	this work	Lerner et al*, Nat. Commun. 2023, 14, 937	Leaptrot et al**, Nat. Commun. 2019, 10, 985
Instrument platform	Trapped ion mobility	Trapped ion mobility	Drift Tube ion mobility
LPE 16:0, CCS [M-H] ⁻ (Å ²)	209.5	210.5	No report
LPE 18:0, CCS [M-H] ⁻ (Å ²)	216.9	218.1	No report
PE 34:1, CCS [M-H] ⁻ (Å ²)	268.6	268.5	266.9
PE 36:2, CCS [M-H] ⁻ (Å ²)	273.8	273.5	272.1
PE 38:4, CCS [M-H] ⁻ (Å ²)	278.1	277.4	276.0
PE O-36:2, CCS [M-H] ⁻ (Å ²)	274.3	274.3	269.0

PE O-38:4 , CCS [M-H] ⁻ (Å ²)	278.2	277.6	No report
PI 34:1, CCS [M-H] ⁻ (Å ²)	290.5	291.3	286.4
PI 36:2, CCS [M-H] ⁻ (Å ²)	295.5	295.4	292.0
PI 38:4 , CCS [M-H] ⁻ (Å ²)	300.0	299.1	296.4
PG 34:1, CCS [M-H] ⁻ (Å ²)	276.0	275.5	273.1
PG 36:2, CCS [M-H] ⁻ (Å ²)	281.3	282.1	279.5
PS 36:2, CCS [M-H] ⁻ (Å ²)	283.7	283.5	281.8
PS 38:4, CCS [M-H] ⁻ (Å ²)	287.6	287.5	285.3

Ref :

* Lerner R, et al. Four-dimensional trapped ion mobility spectrometry lipidomics for high throughput clinical profiling of human blood samples. Nat. Commun. 14, 937 (2023).

**Leaprot KL, May JC, Dodds JN, McLean JA. Ion mobility conformational lipid atlas for high confidence lipidomics. Nat. Commun. 10, 985 (2019).

Page 16, the following sentences are added:

The use of TIMS allowed us to extract the CCS values of 228 phospholipids at the sum composition level based on the measured $1/K_0$ values from bovine liver in negative mode (Supplementary Data 1 and Fig. 3i-k)⁵⁴. These data are consistent with those obtained from other reports^{14, 18} (Supplementary Table 2). Fig. 3i-k show the CCS vs. m/z plots of the identified lipids. Each lipid class is situated in a discrete location, reflecting a significant impact of the headgroup on ion mobility..

Page 16, the following discussion is added:

It should be emphasized that the identified PnEs form distinct trend line in the CCS vs. m/z plot (Fig. 3j, k). PnEs are on average 0.5 \AA^2 smaller than the PEs sharing the same acyl chain composition while 1 \AA^2 larger than the corresponding isomeric PE-O (Fig. 3k). These findings further support that PnE is a distinct subclass of phospholipids and showcase that CCS values can be used as a descriptor to enhance lipid identification.

4. Please ensure that the software (LipidNovelist) and databases used in this study are made freely-available. It does appear that the authors have provided these as part of their submission.

Response: The software and database are freely available. In Code Availability, it is stated: “The LipidNovelist & LipidNovelist Extension, along with instructional videos and example data to facilitate its utilization, can be freely accessed at <https://doi.org/10.6084/m9.figshare.22297771>. ”

5. Some of the authors’ findings clearly point to the impurity/unavailability of current commercial lipid standards (e.g., page 7, line 142; page 15, line 334). This is less of an issue for this work and more of an important issue that should be raised more often whenever possible. The comment here is one of appreciation that the authors are helping to bring the inadequacies of currently-available lipid standards to light.

Response: We appreciate the comment!

6. Other minor issues of note:

(Page 3, line 66) “...by adding added...” can be revised to “...by including...”.

Change: Revision is made as suggested.

(Page 6, line 133) “...separates of...” can be just “...separates...”.

Change: This mistake is corrected.

(Page 7, line 160) “...time needed of...” can be “...time needed for...”

Change: Revision is made as suggested.

(Page 9, line 205, line 207) The use of “C=Cs” is confusing. Recommend “C=C bonds” here.

Change: Revision is made as suggested.

(Page 15, line 332) “...as well as together...”

Change: This mistake is corrected.

(Page 18, line 388) The use of “Rel% n-9” is somewhat confusing. It is repeated enough to warrant this abbreviation, but perhaps introduce it on first use, e.g., “The relative abundance of the n-9 isomer (rel% n-9)...”.

Change: Revision is made accordingly.

(Page 20, line 432) “RAW”

Change: This mistake is corrected.

Reviewer #4

“...Thus, I would strongly advise to publish the manuscript as it is important step in technology development (although some parts are already published by others as well) but after major revision.”

My main concern is in the current form manuscript is rather poorly structured and hard to follow. It is not written in enough details for bioanalytically oriented readers, but also does not have real biological insights which can be important for more biomedical oriented audience. Authors needs to decide which audience they would like to target primary with this work.

Response/Change: We highly appreciate the reviewer’s suggestion. The manuscript has been revised in many places to clarify the key components and importance for clinical applications.

Below are some specific points which might be useful for manuscript revision:

1. Unique point of methodology should be mentioned in the abstract; otherwise, it is rather generic and unattractive.

Response/Change: Thanks for the suggestion. The Abstract has been revised to emphasize the unique analytical advantages of developed method.

2. General proof reading is needed – some sentences are not finished, e.g. p7, line 152, and others.

Response/Change: Thanks for the suggestion. We have carefully edited the whole manuscript.

3. If the method is based on DDA (classical one, without inclusion list), how then particular charged forms (adducts or deprotonated ions) are specifically selected for MS/MS events?

Response: Ionization of phospholipids in negative ion mode via ESI is predictable. NH_4HCO_3 in the LC mobile phase only forms complex ions with lipids containing phosphocholine headgroup, viz. PC, LPC, and SM. Under this condition, all the other classes of phospholipids are ionized as deprotonated ions ($[\text{M} - \text{H}]^-$). Therefore, the classical DDA can be used.

Change: Page 11, the following sentences are added:

“Under the LC-MS condition used, deprotonated ions ($[\text{M}-\text{H}]^-$) are the predominant ionic form for all classes of phospholipids expect for LPC, PC, and SM. For the latter ones the bicarbonate anion adducts $[\text{M} + \text{HCO}_3]^-$ are formed abundantly.”

4. 100 uL of SPLASH for 50 mg of tissue seems really a lot to me... Why such high amount of internal standards was selected? Is it still in the dynamic range of the internal lipids which were quantified relative to the SPLASH standards?

Response: The protocol provided by Avanti suggests an addition of 10 μL of SPLASH into 10 μL of plasma. Based on this, we optimized the quantity of SPLASH for 50 mg of tissue and determined

that an addition of 100 μ L of SPLASH was appropriate. This amount provided a dynamic range of 0.05-15 for relative quantitation (I/I_S) of most classes of phospholipids in bladder tissue.

Change: Page 20, the following sentence is added:

SPLASH (100 μ L) was added to 50 mg tissue before lipid extraction, which provided a dynamic range of 0.05-15 for relative quantitation (I/I_S) of each lipid class.

5. How many cells were used for the experiments?

Response/change: A total of 2 million cells were used for lipid extraction and each LC injection corresponds roughly to \sim 20,000 cells. This information is added to main text and supplementary information.

Page 13, the following sentences are added:

“The performance of the system was further evaluated for profiling unknown phospholipidomes using RAW 264.7 cells (\sim 20,000 cells per injection).

In supplementary information, the following sentences are added:

“A total of 2 million cells were used for lipid extraction”.

6. It is claimed that method take 5 min (even one HILIC run was actually 6 min), but in reality the workflow actually takes much longer. It starts with FA saponification, derivatization and RPLC-MS analysis (7 min) of free FA to generate sample specific fatty acid database. TMSD derivatization and separate of Me PLs adds another 8 min HILIC run, as well as RPLC-MS/MS for TG identification (10 min) when needed.

Response/change: Thanks for pointing this out. The overall analysis time is much more than a single LC run. We have removed this claim and revised the figure for the workflow to correctly represent the analysis time needed for each step. The revised Fig. 2 is shown in the response to question 8.

7. Then also PB derivatization of total lipid extract before HILIC-MS/MS. **And what about combination of pos and neg ion modes?**

Response/change: Thanks for the suggestion. Negative mode MS^2 CID of the PB derivatized phospholipids doesn't provide C=C location information but can provide acyl chain related fragment ions. Given that the chain information can be obtained more sensitively by -ve DDA of the intact phospholipids, we choose to obtain chain information in negative ion mode on intact lipids and C=C location information in positive ion mode on the PB derivatized products.

Change: Page 9, the following sentence is added.

“...Note that PB- MS^2 CID of phospholipids in negative ion mode is much less sensitive in providing detailed structural information.”

8. Detailed scheme representing the whole workflow with sequence of steps and brief explanations is needed to support readability of the manuscript. **Figure 2 does not reflect the real composition of the methodology and thus very misleading.**

Response/change: We have revised Figure 2 and related description of the analytical procedures (Page 11).

9. 3 provided application examples create more confusions than clarity.... In each of them different lipids are reported at the different levels of structural annotations, results are not really discussed appropriately from biological or biomedical perspectives, so readers will be left with the feeling that methodology produce a lot of data but usability of these data remains mostly unclear. I would suggest to focus on one example (and if wanted, move the other 2 into supplementary information) but present it in much more details and clarity, starting with data acquisition strategy ending up with discussion of biological significance.

Response: Thanks for the suggestion. We may not clearly articulate the logic behind the three examples (bovine liver, RAW cell line, bladder cancer tissue); however, they are indispensable to demonstrate the performance and potential application of the developed deep profiling system for the following reasons:

1. The analysis of polar lipid extract of bovine liver was used to benchmark the performance of the HILIC-TIMS-isomer-resolved MS/MS system, which was shown to have many analytical advantages over the previously developed HILIC-PB-MS/MS system (Zhang et al., 2019, Nat. Commun.).
2. The RAW 264.7 cell line was used to demonstrate how the developed system can be applied to profile the phospholipidome of an unknown sample. We established a sample-

specific FA database by correlating desaturase inhibition and the change of C=C location isomers in total FAs. This method successfully overcomes the practical limitation of obtaining lipid C=C isomer standards and greatly improved the confidence for phospholipid annotation.

3. The human bladder cancer tissue was to showcase that relative quantitation of the lipid isomers (C=C, PC sn) enables phenotyping of cancerous tissue at the molecular level, which could not be achieved by traditional profiling methods.

Change: We have revised the discussions in the main text (Page 13) to emphasize the logic behind the designed experiments and associated novelty.

“For benchmark purpose, the above system was applied to analyze the polar lipid extract of bovine liver (1 µg lipid extracts per injection), a commercially available lipid mixture with phospholipids being extensively analyzed at the C=C location level using different PB-MS/MS methods^{36,46}. The performance of the system was further evaluated for profiling unknown phospholipidomes using RAW 264.7 cells (~20,000 cells per injection). A significant aspect of this study was the development of a sample-specific TFA database by correlating desaturase inhibition with the compositional changes of C=C isomers in TFAs. This approach overcame the practical limitation of obtaining lipid C=C isomer standards and significantly improved the confidence in phospholipid annotation. Finally, to demonstrate the unique capability of deep profiling in clinical applications, the human bladder tissue samples were analyzed (100 µg tissue per injection).”

REVIEWERS' COMMENTS

Reviewer #1 (Remarks to the Author):

It is my pleasure to write that the authors have performed the revision in a responsible manner and responded to all of my comments in the way that I am certified with all answers and can recommend the publication of their work. I would like to emphasize that the changes made a significant improvement for some issues in the previous version. I appreciate the detailed reasoning for the novelty of their work compared to what they already published and the changes made in the manuscript. I have noticed that some other reviewers made similar comments. However, I am satisfied with the explanation of novelty now. The issue of quantitation is more complicated. I know the range of quantitative standards available on the market, so the use of the SPLASH mix is fine for me. Previously, it was not clear to me whether the authors used any internal standard at all, which is clarified now and is also reflected in the manuscript text. All other responses are satisfactory to me and do not require further comments from my side.

Reviewer #2 (Remarks to the Author):

The authors have addressed the technical points I raised in my original review and have clarified key aspects on what the new workflow represents. Figure 2 is helpful in this regard showing that there are indeed multiple parallel analyses to arrive at a more comprehensive description of the lipidome. In this respect, it does confirm my assessment that these aspects have been previously published (mostly by this team) and this report describes how the aspects from each prior contribution can be combined for more robust lipid assignment and thus a more comprehensive lipidome description. The benchmark reported (and to some extent the key claim to novelty) is a doubling of their own previous number of assignments for bovine liver extract; rather than a comparison of the number of lipids identified in the well-studied RAW 264 cell lines. While the current study clearly provides a deeper dive into the glycerophospholipids there are a significant number of lipid classes previously studied in these cell lines that are not reported here. My assessment remains therefore that this is a significant but incremental advance.

As a minor comment, in their responses to my comments regarding the missed opportunity to exploit the ion mobility for isomer separation. In their assessment, the authors put some stock in the CCS values for isomers taken from Zhou et al. (10.1021/acs.analchem.7b02625). It should be remembered that these are machine learning predictions not measurements – as the authors note themselves there have up until recently been few available reference standards but sn-isomers for PC 16:0_20:4 and LPC 16:1 etc. are now available and it would be valuable to the field to assess their mobility behaviours in this pipeline.

Reviewer #3 (Remarks to the Author):

The authors have addressed the primary concerns of this reviewer (#3) and with the edits/additions made to the text, this work is now recommended for publication.

Reviewer #4 (Remarks to the Author):

In revised version of the manuscript authors addressed all points of concerns from my previous revision. Current version and updated figure 2 provides now much more realistic reflection on the proposed workflow. Other critical points were addressed and corrected accordingly. I recommend to accept the manuscript for the publication in its current form.

Point-by-Point response to reviewers' comments

Reviewer #1 (Remarks to the Author):

It is my pleasure to write that the authors have performed the revision in a responsible manner and responded to all of my comments in the way that I am certified with all answers and can recommend the publication of their work. I would like to emphasize that the changes made a significant improvement for some issues in the previous version. I appreciate the detailed reasoning for the novelty of their work compared to what they already published and the changes made in the manuscript. I have noticed that some other reviewers made similar comments. However, I am satisfied with the explanation of novelty now. The issue of quantitation is more complicated. I know the range of quantitative standards available on the market, so the use of the SPLASH mix is fine for me. Previously, it was not clear to me whether the authors used any internal standard at all, which is clarified now and is also reflected in the manuscript text. All other responses are satisfactory to me and do not require further comments from my side.

Response: No further revision is needed from this reviewer.

Reviewer #2 (Remarks to the Author):

The authors have addressed the technical points I raised in my original review and have clarified key aspects on what the new workflow represents. Figure 2 is helpful in this regard showing that there are indeed multiple parallel analyses to arrive at a more comprehensive description of the lipidome. In this respect, it does confirm my assessment that these aspects have been previously published (mostly by this team) and this report describes how the aspects from each prior contribution can be combined for more robust lipid assignment and thus a more comprehensive lipidome description. The benchmark reported (and to some extent the key claim to novelty) is a doubling of their own previous number of assignments for bovine liver extract; rather than a comparison of the number of lipids identified in the well-studied RAW 264 cell lines. While the current study clearly provides a deeper dive into the glycerophospholipids there are a significant number of lipid classes previously studied in these cell lines that are not reported here. My assessment remains therefore that this is a significant but incremental advance.

As a minor comment, in their responses to my comments regarding the missed opportunity to exploit the ion mobility for isomer separation. In their assessment, the authors put some stock in the CCS values for isomers taken from Zhou et al. (10.1021/acs.analchem.7b02625). It should be remembered that these are machine learning predictions not measurements – as the authors note themselves there have up until recently been few available reference standards but sn-isomers for PC 16:0_20:4 and LPC 16:1 etc. are now available and it would be valuable to the field to assess their mobility behaviours in this pipeline.

Response/Change: The reviewer is correct that one should be cautious in referring to CCS values generated by AI (Zhu lab) because they are largely not calibrated by experimental measurements. We thus replaced this reference with Groessl et al. (Analyst, 2015, 140, 6904), in which the CCS values of various metal adduct ions of PC 16:0/18:1 and PC 18:1/16:0 were reported (listed below).

Editorial Note: [redacted]

Reviewer #3 (Remarks to the Author):

The authors have addressed the primary concerns of this reviewer (#3) and with the edits/additions made to the text, this work is now recommended for publication.

Response: No further revision is needed from this reviewer.

Reviewer #4 (Remarks to the Author):

In revised version of the manuscript authors addressed all points of concerns from my previous revision. Current version and updated figure 2 provides now much more realistic reflection on the proposed workflow. Other critical points were addressed and corrected accordingly. I recommend to accept the manuscript for the publication in its current form.

Response: No further revision is needed from this reviewer.